# SINCERE: Supervised Information Noise-Contrastive Estimation REvisited

## Abstract

The information noise-contrastive estimation (InfoNCE) loss function provides the basis of many self-supervised deep learning methods due to its strong empirical results and theoretic motivation. Previous work suggests a supervised contrastive (SupCon) loss to extend InfoNCE to learn from available class labels. This SupCon loss has been widely-used due to reports of good empirical performance. However, in this work we find that the prior SupCon loss formulation has questionable justification because it can encourage some images from the same class to repel one another in the learned embedding space. This problematic intra-class repulsion gets worse as the number of images sharing one class label increases. We propose the Supervised InfoNCE REvisited (SINCERE) loss as a theoretically-justified supervised extension of InfoNCE that eliminates intra-class repulsion. Experiments show that SINCERE leads to better separation of embeddings from different classes and improves transfer learning classification accuracy. We additionally utilize probabilistic modeling to derive an information-theoretic bound that relates SINCERE loss to the symmeterized KL divergence between data-generating distributions for a target class and all other classes.

## 1 Introduction

Self-supervised learning (SSL) has been crucial in creating pretrained computer vision models that can be efficiently adapted to a variety of tasks (Jing & Tian, 2020; Jaiswal et al., 2021). The conceptual basis for many successful SSL methods is the instance discrimination task (Wu et al., 2018), where the model learns to classify each training image as a unique class. Self-supervised methods solve this task by *contrasting* different augmentations of the same image with other images, seeking a learned vector representation in which each image is close to augmentations of itself but far from others. Among several possible contrastive losses in the literature (Caron et al., 2020; Schroff et al., 2015), one that has seen particularly wide adoption is information noise-contrastive estimation (InfoNCE) (van den Oord et al., 2018). InfoNCE variants such as MoCo (Chen et al., 2021), SimCLR (Chen et al., 2020a;b), and BYOL (Grill et al., 2020) have proven empirically effective.

The aforementioned methods are all for *self-supervised* pretraining of representations from unlabeled images. Instance discrimination methods may be extended for *supervised* applications to learn representations informed by the available class labels. A natural way forward is to contrast images of the same class with images from other classes (Schroff et al., 2015). The noise contrastive estimation framework (Gutmann & Hyvärinen, 2010) implements this idea by assuming that images from the same class are drawn from a target distribution while images from other classes come from a noise distribution.

Khosla et al. (2020) proposed the supervised contrastive (SupCon) loss as a supervised extension of the InfoNCE loss, forming target and noise distributions using supervised labels. They sought a loss that could achieve the following goal: "Clusters of points belonging to the same class are pulled together in embedding space, while simultaneously pushing apart clusters of samples from different classes." Khosla et al. (2020)'s recommended loss, named SupCon, was chosen because it performed best empirically in terms of classification accuracy. In followup work, SupCon loss has been applied to problems such as contrastive open set recognition (Xu et al., 2023) and generalized category discovery (Vaze et al., 2022).

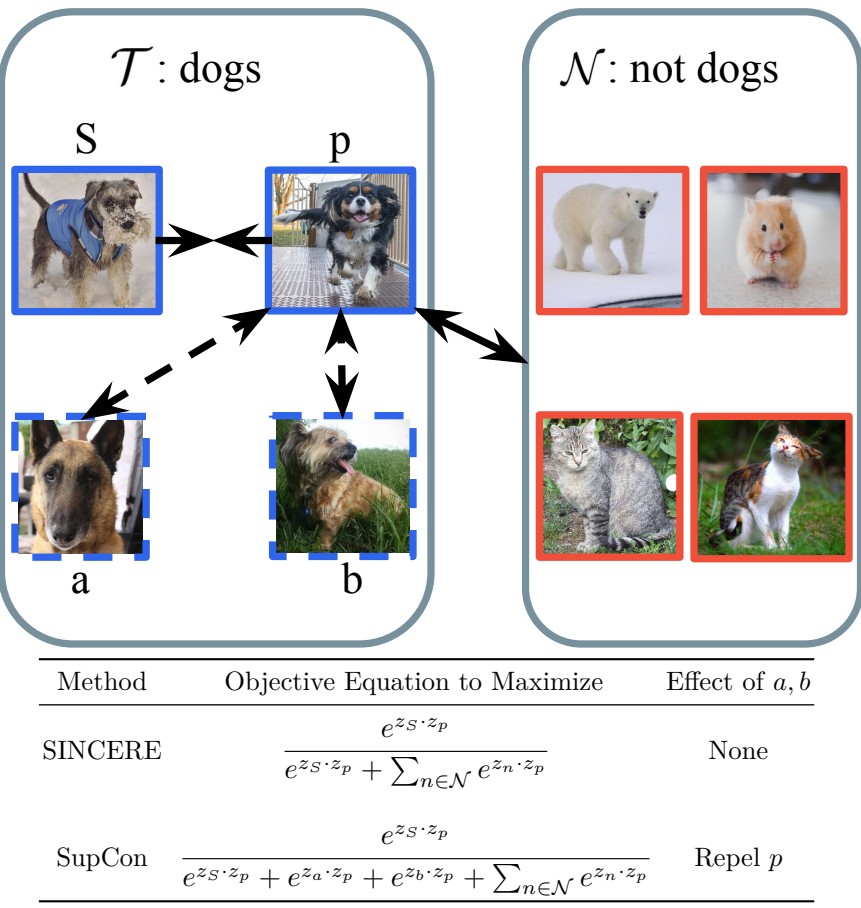

Figure 1: Visualization of supervised contrastive learning objectives for pulling together embeddings from the target class, indexed by elements of $\mathcal{T}$, and pushing away embeddings from the noise classes, indexed by elements of $\mathcal{N}$. Both objectives are defined with respect to a pair of target embeddings $z_S$ and $z_p$. Solid arrows show *common* effects of both methods: $z_p$ is pulled towards $z_S$ and pushed away from embeddings $z_n$ from the noise classes. Dashed arrows show *SupCon's problematic intra-class repulsion*: $z_p$ is pushed away from $z_a$ and $z_b$ as if they were from a noise class, despite belonging to the target class.

In light of this empirical success, we investigate the theoretical justification for SupCon in this work. We find that SupCon does not separate target and noise distributions as well as possible due to inconsistent distribution definitions producing *intra-class repulsion*. Consider the situation illustrated in Fig. 1: we have a target class "dog" with at least 3 member images in the current batch, referred to as $S$, $p$, and $a$. The SupCon objective is defined as an average over many pairs representing the target class. When the current pair of interest is target image $S$ and same-class partner $p$, optimizing SupCon pulls embeddings of $S$ and $p$ together, but problematically treats dog image $a$ as a noise image and pushes $p$'s representation away from $a$. This inconsistency makes it difficult to separate target and noise classes in the embedding space. Moreover, the problem gets worse as the number of target-class images increases, causing more target images to be treated as noise. Surprisingly, neither the original work on SupCon nor variants of SupCon (Kang et al., 2021; Feng et al., 2022; Li et al., 2022b; Barbano et al., 2023) have characterized this intra-class repulsion problem.

To resolve this issue, this paper proposes the Supervised InfoNCE REvisited (SINCERE) loss. When evaluated on the scenario in Fig. 1 with image pair $S, p$ representing the target class, our SINCERE loss excludes other target images such as $a$ from the noise distribution, unlike SupCon. This choice ensures *consistent* definitions of target and noise distributions as in the original self-supervised InfoNCE, thereby eliminating intra-class repulsion. Other recently proposed losses for self-supervised learning (Chuang et al., 2020; Yeh et al., 2022), discussed in Sec. 3.5, also make changes to the denominator of an InfoNCE-like loss, but focus on other

problems such as correcting a "sampling bias" in the noise distribution or improving efficiency by removing a "coupling" effect. Their ultimate losses are not intended for *supervised* tasks.

Overall, our main contributions are:

1. We identify intra-class repulsion as a key issue with SupCon loss, demonstrating its problematic effects through a formal analysis of gradients (Sec. 3.4) and an empirical finding of notably smaller target-noise repulsion in the learned embeddings (see thin margin of separation for SupCon in Fig. 2).

2. We propose the SINCERE loss function for supervised representation learning as a drop-in replacement for SupCon that eliminates intra-class repulsion. We derive SINCERE via a first-principles analysis of a probabilistic generative model while enforcing a core assumption of noise-contrastive estimation: images known to be from target distribution should not be treated as noise examples. SINCERE is thus a well-founded generalization of InfoNCE that can use available class labels. Interestingly, we show that SINCERE corresponds to the $\epsilon$-SupInfoNCE loss of Barbano et al. (2023) when $\epsilon = 0$. Barbano et al. did not raise the intra-class repulsion issue, used geometric rather than probabilistic arguments to motivate their loss, and recommend $\epsilon > 0$ values requiring expensive hyperparameter tuning without comparing to the simpler case of $\epsilon = 0$.

3. We prove in Thm. 11 that our idealized SINCERE loss acts as an upper bound on the negative symmeterized KL divergence between the data-generating target and noise distributions. This information-theoretic analysis sheds light on how the number of noise samples and the similarity of target and noise distributions impact loss values.

4. We show in our empirical results that SINCERE loss eliminates the problematic intra-class repulsion behavior of the SupCon loss (see large margin of separation for SINCERE in Fig. 2). This leads to superior accuracy for transfer learning (see Tab. 2) over both SupCon and $\epsilon$-SupInfoNCE (Barbano et al., 2023) in 4 out of 6 datasets, even when that latter method is allowed to tune an extra hyperparameter $\epsilon > 0$.

Ultimately, practitioners that use SINCERE can enjoy the same or better downstream classification accuracy as SupCon while benefiting from a solid conceptual foundation and *improved separation* of target and noise distributions in the learned embedding space. Code for reproducing all experiments is available in supplementary materials.

## 2 Background

To begin, we introduce notation used throughout this paper. See also the notation summary in Appendix Table 3 for easy reference. Consider an observed data set $(\mathcal{X}, \mathcal{Y})$ of $N$ elements. Let $\mathcal{X} = (x_1, x_2, ..., x_N)$ define the data feature vectors (e.g. images), and $\mathcal{Y} = (y_1, y_2, ..., y_N)$ the categorical labels. Each example (indexed by integer $i$) has a feature vector $x_i$ paired with an integer-valued categorical label $y_i \in [\![1, K]\!]$, where $K$ indicates the number of classes and $2 \leq K \leq N$. Let integer interval $\mathcal{I} = [\![1, N]\!]$ denote the set of possible indices for elements in $\mathcal{X}$ or $\mathcal{Y}$.

### 2.1 Noise-Contrastive Estimation

Noise-contrastive estimation (NCE) (Gutmann & Hyvärinen, 2010) provides a framework for modeling a target distribution of interest given a set of samples from it. This framework utilizes a binary classifier to contrast the target distribution samples with samples from a noise distribution. This noise distribution is an arbitrary distribution different from the target distribution, although in practice the noise distribution must be similar enough to the target distribution to make the classifier learn the structure of the target distribution (Gutmann & Hyvärinen, 2010). Works described in subsequent sections maintain focus on contrasting target and noise distributions while defining these distributions as generating disjoint subsets of a data set.

## 2.2 Self-Supervised Contrastive Learning

Self-supervised contrastive learning pursues an *instance discrimination* task (Wu et al., 2018) to learn effective representations. This involves treating each example in the data set as a separate class. Therefore each example $i$ has a *unique* label $y_i$ and the number of classes $K$ is equal to $N$.

To set up the instance discrimination problem, let $S \in \mathcal{I}$ denote the index of the only member of the target distribution in the data set. Let the rest of the data set $\mathcal{N} = \mathcal{I} \setminus \{S\}$ be drawn from the noise distribution. Applying NCE produces the Information Noise-Contrastive Estimation (InfoNCE) loss (van den Oord et al., 2018)

$$L_{\text{InfoNCE}}(x_S, y_S) = -\log \frac{e^{f(x_S, y_S)}}{e^{f(x_S, y_S)} + \sum_{n \in \mathcal{N}} e^{f(x_n, y_S)}} \tag{1}$$

where $f(x_i, y_j)$ is a classification score function which outputs a scalar score for image $x_i$ where greater values indicate that label $y_j$ is more likely. When $y_S$ denotes the target class, loss $L_{\text{InfoNCE}}(x_S, y_S)$ thus calculates the negative log-likelihood that index $S$ correctly locates the only sample from the target class.

The function $f$ is often chosen to be cosine similarity by representing both terms as vectors in an embedding space (Wu et al., 2018; Chen et al., 2020a; 2021). Let $z_i \in \mathbb{R}^D$ be a $D$-dimensional unit vector representation of data element $x_i$ produced by a neural network. Embeddings are also used to represent the target instance label $y_S$, so let $z'_S$ be a second representation of the target instance $x_S$. $z'_S$ can be produced by embedding a data augmented copy of $x_S$ (Le-Khac et al., 2020; Chen et al., 2020a), embedding $x_S$ via older (Wu et al., 2018) or averaged (Chen et al., 2021) embedding function parameters, or a combination of these techniques (Jaiswal et al., 2021).

Rewriting $L_{\text{InfoNCE}}$ in terms of a data augmented $z'_S$ and setting $f(z_i, z_j) = z_i \cdot z_j / \tau$, with $\tau > 0$ acting as a temperature hyperparameter, produces the self-supervised contrastive loss proposed by Wu et al. (2018)

$$L_{\text{self}}(z_S, z'_S) = -\log \frac{e^{z_S \cdot z'_S / \tau}}{e^{z_S \cdot z'_S / \tau} + \sum_{n \in \mathcal{N}} e^{z_n \cdot z'_S / \tau}} \tag{2}$$

Note that we assume each embedding vector is a unit vector ($z_i \cdot z_i = 1$ for all $i$), so the dot products in the equation above will always be in $[-1, 1]$ before scaling by $\tau$.

InfoNCE and the subsequent self-supervised contrastive losses cited previously are all theoretically motivated by NCE, as described in van den Oord et al. (2018). The larger instance discrimination problem is posed as a series of binary classification problems between instance-specific target and noise distributions. This clear distinction between target and noise underlies our later SINCERE loss.

## 2.3 Supervised Contrastive Learning (SupCon)

In the supervised setting, labels $y_i$ are no longer unique for each data point. Instead, we assume a fixed set of $K$ known classes, such as "dogs" and "cats," with multiple examples in each class. Again, the larger $K$-way discrimination task is posed as a series of binary NCE tasks. Each binary task distinguishes one target class from a noise distribution made up of the $K - 1$ remaining classes.

Let index $S$ again denote a selected instance that defines the current target class $y_S$. Let $\mathcal{T} = \{i \in \mathcal{I} | y_i = y_S\}$ be the set of all elements from the target class: unlike the self-supervised case, here $\mathcal{T}$ has *multiple* elements. Let the possible set of same-class partners for index $S$ be $\mathcal{P} = \mathcal{T} \setminus \{S\}$. Let the set of indices from the noise distribution be $\mathcal{N} = \mathcal{I} \setminus \mathcal{T}$.

Khosla et al. (2020) propose a supervised contrastive loss known as "SupCon". For chosen $S$, the overall loss is $\frac{1}{|\mathcal{P}|} \sum_{p \in \mathcal{P}} L_{\text{SupCon}}(z_S, z_p)$, with the pair-specific loss $L_{\text{SupCon}}$ defined as

$$L_{\text{SupCon}}(z_S, z_p) = -\log \frac{e^{z_S \cdot z_p / \tau}}{e^{z_S \cdot z_p / \tau} + \sum_{j \in \mathcal{P} \setminus \{p\}} (e^{z_j \cdot z_p / \tau}) + \sum_{n \in \mathcal{N}} (e^{z_n \cdot z_p / \tau})} \tag{3}$$

where $z_p$ denotes another embedding from the target class. Though our notation differs, the above is equivalent to the overall loss in Equation 2 of the original SupCon work by Khosla et al. (2020). See App. I for details.

Khosla et al. (2020) suggest this loss as a "straightforward" extension of $L_{\text{self}}$ to the supervised case, with the primary justification given via their reported empirical success on classification, especially on the widely-used ImageNet data set (Deng et al., 2009).

### 2.3.1 Intra-Class Repulsion

The core issue with SupCon motivating our work is *intra-class repulsion*. Despite SupCon's stated goal that images of the same class "are pulled together in embedding space" (Khosla et al., 2020), we find that some image pairs sharing a label can be pushed apart in embedding space. Examining equation 3, we notice the SupCon loss contains terms from the target distribution in the denominator, indexed by $j$ when $j \neq S$, that are not in the numerator when $|\mathcal{P}| > 1$. In contrast, the self-supervised loss in equation 2 includes all target terms in both numerator and denominator. SupCon's choice to have some target examples only in the denominator of equation 3 effectively treats them as part of the noise distribution. That is, the SupCon loss will favor pushing such embeddings $z_j$ away from $z_p$, as illustrated in Fig. 1. This problematic intra-class repulsion complicates analysis of the loss (Graf et al., 2021) and limits SupCon's ability to achieve its stated goal: separate embeddings by class.

## 3 Method

In this section, we develop a revised loss for supervised contrastive learning called SINCERE. We derive and justify our SINCERE loss in Sec. 3.1, showing how it arises from applying noise-contrastive estimation to a supervised problem via the same probabilistic principles that justify InfoNCE for self-supervised learning. In fact, InfoNCE becomes a special case of our proposed SINCERE framework. Sec. 3.2 derives an information-theoretic bound on the SINCERE loss. Sec. 3.3 describes a practical implementation of the SINCERE loss. Sec. 3.4 shows the SINCERE loss gradient eliminates the intra-class repulsion present in the SupCon loss gradient. Finally, Sec. 3.5 examines how SINCERE loss relates to other works building on InfoNCE and SupCon losses, especially highlighting SINCERE's unexpected correspondence to $\epsilon$-SupInfoNCE of Barbano et al. (2023), a loss motivated by geometric arguments without any probabilistic analysis.

### 3.1 Derivation of SINCERE

We first review the principles that establish InfoNCE loss as suitable for self-supervised learning. We then extend that derivation to the more general supervised learning case. Each derivation proceeds in three steps. First, we establish a data-generating probabilistic model involving a target and a noise distribution. Second, we formulate a one-from-many selection task: given a batch of many images drawn from the model, we must select which one index comes from the target distribution. Third, we pursue a tractable model for this selection task, parameterized by a neural network, for the applied case when the data-generating distributions are not known. Ultimately, our proposed SINCERE loss, like InfoNCE before it, is interpreted as a negative log likelihood of the tractable neural network approximation for this selection task.

### 3.1.1 Self-Supervised Probabilistic Model

In the self-supervised case, instance discrimination is posed as a series of binary target-noise discrimination tasks. Each target distribution produces images that depict a specific instance, such as data augmentations of a single image. The corresponding noise distribution generates images of other possible instances. Throughout this subsection, we focus analysis on *just one* of these binary tasks, treating its target and noise distributions as fixed. Later, Sec. 3.3 will describe how our approach models multiple target-noise tasks in practice.

**Assumption 1.** *We observe a data set $\mathcal{X}$ of $N$ examples with unknown labels such that exactly one example's data is drawn from the target distribution.*

Let random variable $S \in \mathcal{I}$ indicate the index of the example sampled from the target distribution with PDF $p^+(x_i)$. All other examples $\mathcal{N} = \mathcal{I} \setminus \{S\}$ are drawn i.i.d. from the noise distribution with data-generating

PDF $p^-(x_i)$. We assume $p^-(x_i)$ is nonzero whenever $p^+(x_i)$ is nonzero, as in Gutmann & Hyvärinen (2010). Unlike the alternative generative model in Arora et al. (2019), we do not require the noise distribution to be a mixture of target and non-target latent classes; instead we merely assume the noise distribution is distinct from the target, without any need for latent classes.

**Definition 2.** *The "true" data-generating model for the self-supervised case, under the above assumptions, generates index $S$ and then data set $\mathcal{X}$ from the distributions below*

$$p(\mathcal{X}, S) = p(\mathcal{X}|S)p(S), \qquad p(S) = \mathit{Unif}(\mathcal{I}) \tag{4}$$
$$p(\mathcal{X}|S) = p^+(x_S) \prod_{n \in \mathcal{N}} p^-(x_n)$$

*where $\mathit{Unif}(\mathcal{I})$ is the uniform distribution over indices from 1 to $N$. The assumed model thus factorizes as $p(\mathcal{X}, S) = p(\mathcal{X}|S)p(S)$.*

This model is used to solve a selection task: given data set $\mathcal{X}$, which single index is from the target class?

**Proposition 3.** *Assume $\mathcal{X}$ is generated via the model in Def. 2 and that the target and noise PDFs are known. The probability that index $S$ is the sole draw from the target distribution is*

$$p(S|\mathcal{X}) = \frac{p^+(x_S) \prod_{n \in \mathcal{N}} p^-(x_n)}{\sum_{i \in \mathcal{I}} p^+(x_i) \prod_{j \in \mathcal{I} \setminus \{i\}} p^-(x_j)} \tag{5}$$

$$= \frac{\frac{p^+(x_S)}{p^-(x_S)}}{\frac{p^+(x_S)}{p^-(x_S)} + \sum_{n \in \mathcal{N}} \frac{p^+(x_n)}{p^-(x_n)}}. \tag{6}$$

**Proof** Bayes theorem produces the first formula given the joint $p(\mathcal{X}, S)$ defined in equation 4. Algebraic simplifications lead to the second formula, recalling $\mathcal{I} = \mathcal{N} \cup \{S\}$. ∎

This proposition formalizes a result from van den Oord et al. (2018) used to motivate their InfoNCE loss. As argued formally in Sec. 3.1.4, a neural network $f_\theta$ that learns to approximate $\frac{p^+(x_i)}{p^-(x_i)}$ minimizes InfoNCE loss.

### 3.1.2 Supervised Probabilistic Model

In the supervised case, supervised classification is posed as a series of binary target-noise discrimination tasks. Each target distribution produces images that depict a single class, such as "dog." The corresponding noise distribution generates images of all other classes, such as "cat" and "hamster." As above, throughout this subsection we assume one fixed target and one fixed noise distribution. Later sections will describe how to attack multiple target-noise tasks in practice.

**Assumption 4.** *We observe a data set $\mathcal{X}$ of $N$ examples with unknown labels such that exactly $T$ examples represent draws from the target distribution, with $2 \leq T < N$.*

Let random variable $\mathcal{P} \in \{I \subset \mathcal{I} \mid |I| = T-1\}$ indicate a set of $T-1$ indices identifying all but one of the examples from the target distribution with data-generating PDF $p^+(x_i)$. Let random variable $S \in \mathcal{I} \setminus \mathcal{P}$ indicate the index of the final example sampled from the target distribution. The set of all indices from the target distribution is denoted $\mathcal{T} = \mathcal{P} \cup \{S\}$. The remaining indices $\mathcal{N} = \mathcal{I} \setminus \mathcal{T}$ are drawn i.i.d. from the noise distribution, whose PDF is $p^-(x_i)$.

**Definition 5.** *The "true" data-generating model for the supervised case is*

$$p(\mathcal{X}, S, \mathcal{P}) = p(\mathcal{X}|S, \mathcal{P})p(S|\mathcal{P})p(\mathcal{P}), \qquad p(\mathcal{P}) = \mathit{Unif}(\{I \subset \mathcal{I} \mid |I| = T-1\}) \tag{7}$$
$$p(S|\mathcal{P}) = \mathit{Unif}(\mathcal{I} \setminus \mathcal{P})$$
$$p(\mathcal{X}|S, \mathcal{P}) = p^+(x_S) \prod_{p \in \mathcal{P}} p^+(x_p) \prod_{n \in \mathcal{N}} p^-(x_n)$$

*where $\mathcal{P}$'s distribution is uniform over sets of exactly $T-1$ distinct indices within the larger set $\mathcal{I}$.*

In the special case of $T = 1$, note that $\mathcal{P}$ becomes the empty set, and the supervised model in equation 7 (where knowledge of $\mathcal{P}$ provides additional information about the target class) reduces to the simpler self-supervised model in equation 4. This connection suggests our SINCERE framework can also be used for instance discrimination with more than two augmentations of each image instance.

The following likelihood of the selection task index given observed $\mathcal{X}$ and $\mathcal{P}$ values generalizes equation 6 to the supervised case and motivates the SINCERE loss.

**Proposition 6.** *Assume $\mathcal{X}$ and $\mathcal{P}$ are generated from the joint distribution defined in equation 7 and that the target and noise PDFs are known. The probability that any index $S$ in $\mathcal{I} \setminus \mathcal{P}$ is the final draw from the target distribution is*

$$p(S|\mathcal{X}, \mathcal{P}) = \frac{\frac{p^+(x_S)}{p^-(x_S)}}{\frac{p^+(x_S)}{p^-(x_S)} + \sum_{n \in \mathcal{N}} \frac{p^+(x_n)}{p^-(x_n)}}. \tag{8}$$

**Proof** We first derive an expression for $p(S, \mathcal{P}|\mathcal{X})$ from the joint defined in equation 7. Standard probability operations (sum rule, product rule) then allow obtaining the desired $p(S|\mathcal{X}, \mathcal{P})$. For details, see App. B. ∎

In contrast to SINCERE, attempts to translate SupCon loss in equation 3 into the noise-contrastive paradigm do *not* result in a coherent probabilistic model, as detailed in App. F. In short, SupCon loss cannot be viewed as a valid PMF for the conditional probability $p(S|\mathcal{X}, \mathcal{P})$ due to extra target class terms in the denominator.

### 3.1.3  Ideal SINCERE Loss

In most applications, we can only observe a (partially) labeled dataset. We will not know the true density functions $p^+$ and $p^-$ for target and noise distributions, as assumed in equation 8. Instead, given only data $\mathcal{X}$ and indices $\mathcal{P}$ that represent the target class, we can build an alternative tractable model for determining the index $S$ of the final member of the target class. Here we define this model and a loss to fit it to data.

**Definition 7.** *Let neural net $f_\theta(x_i, y_S)$, with parameters $\theta$, map any input data $x_i$ and target class $y_S$ to a real value indicating the relative confidence that $x_i$ belongs to the target class, where a greater value implies more confidence. Our tractable model for selecting index $S$ given data set $\mathcal{X}$ and known class members $\mathcal{P}$ is*

$$p_\theta(S|\mathcal{X}, \mathcal{P}) = \frac{e^{f_\theta(x_S, y_S)}}{e^{f_\theta(x_S, y_S)} + \sum_{n \in \mathcal{N}} e^{f_\theta(x_n, y_S)}}. \tag{9}$$

*where by definition, given each possible $S$ we construct the noise index set as $\mathcal{N} = \mathcal{I} \setminus (\mathcal{P} \cup S)$.*

We need a way to estimate the parameters $\theta$ of this tractable model. Suppose we can observe many samples of $\mathcal{X}, S, \mathcal{P}$ from the true generative model in equation 7, even though we lack direct access to the PDF functions $p^+$ and $p^-$. In this setting we can fit $f_\theta$ by minimizing what we call the "ideal" SINCERE loss.

**Definition 8.** *The SINCERE loss for the generative model in equation 7 is defined as*

$$L_{SINCEREideal}(\theta) = \mathbb{E}_{\mathcal{X}, S, \mathcal{P}} \left[ -\log p_\theta(S|\mathcal{X}, \mathcal{P}) \right], \tag{10}$$

*where evaluating the expectation assumes ideal access to many iid samples from the generative model.*

This proposed SINCERE loss provides a principled way to fit a tractable neural model $f_\theta$ to identify the last remaining member of a target class when given a data set $\mathcal{X}$ and other class member indices $\mathcal{P}$. The definition here covers the idealized case where the expectation is over samples drawn independently from the true generative model. This would be prohibitively expensive in practice, so we provide a definition that uses mini-batches of a finite labeled dataset for more efficient learning in Sec. 3.3.

### 3.1.4  Justification for SINCERE Loss

We justify the chosen form of function $f_\theta$ in equation 9 and loss $L_{SINCEREideal}$ in equation 10 in two steps. First, we suggest it is possible to set parameters $\theta$ to a value $\theta^*$ such that the tractable model $p_{\theta^*}(S|\mathcal{X}, \mathcal{P})$

exactly matches the true distribution $p(S|\mathcal{X}, \mathcal{P})$ in equation 8. This step is formalized in Assumption 9 below. Second, we prove this matching parameter $\theta^*$ will be a minimizer of the SINCERE loss. This step is formalized in Theorem 10.

**Assumption 9.** *The function class of neural network $f_\theta$ is sufficiently flexible, such that there exists parameters $\theta^*$ satisfying $e^{f_{\theta^*}(x_i, y_S)} = \frac{p^+(x_i)}{p^-(x_i)}$ for all possible data vectors $x_i$.*

Universal approximation theorems for neural networks (Lu et al., 2017) suggest this function approximation task is achievable. Given a parameter $\theta^*$ meeting this assumption, substituting that in our tractable selection likelihood in equation 10 straightforwardly recovers the true conditional probability in equation 8.

**Theorem 10.** *Parameters $\theta^*$ that satisfy Assumption 9 minimize the SINCERE loss defined in equation 10, where the expectation is over samples of $\mathcal{X}, S, \mathcal{P}$ from the generative model equation 7.*

**Proof** The loss minimization objective equation 10 is equivalent to maximizing the tractable log likelihood $p_\theta(S|\mathcal{X}, \mathcal{P})$ under samples from the generative model. The theory of maximum likelihood estimation under model misspecification (White, 1982; Fan, 2016) shows minimizing equation 10 is equivalent to minimizing the KL-divergence $KL(p(S|\mathcal{X}, \mathcal{P})||p_\theta(S|\mathcal{X}, \mathcal{P}))$, where $p(S|\mathcal{X}, \mathcal{P})$ (without any subscript) denotes the conditional likelihood of the selection index in equation 8 arising from the generative model. KL-divergence is minimized and equal to 0 when its two arguments are equal. Parameter vector $\theta^*$ makes the arguments equal by construction, therefore the minimum of the SINCERE loss occurs at the vector $\theta^*$. ∎

This two-step justification for our proposed SINCERE loss holds both when $\mathcal{P}$ is non-empty, as with SupCon, as well the special case where $\mathcal{P}$ is the empty set, where SINCERE is equivalent to InfoNCE. These steps formalize the arguments for InfoNCE presented by van den Oord et al. (2018) and extend them to handle the more general supervised scenario. Ultimately, this justification shows that minimizing the SINCERE loss is a principled way to fit a tractable model for both the supervised contrastive classification task and the self-supervised instance discrimination task. In contrast, the lack of coherent probabilistic model motivating SupCon, as shown in App. F, precludes an analogous analysis.

### 3.2 Lower Bound on SINCERE Loss

Previous work by van den Oord et al. (2018) motivated the self-supervised InfoNCE loss via an information-theoretic bound related to mutual information. We revisit this analysis for the more general case of SINCERE loss under the idealized settings of Sec. 3.1, where there is one target-noise task of interest, with target PDF $p^+(x_i)$ and noise PDF $p^-(x_i)$. These results generalize to InfoNCE loss with the additional assumption that $T = 1$, that is the data set contains only one sample from the target PDF.

In general, by the definition of loss $L_{\text{SINCEREideal}}(\theta)$ in equation 10 as an expectation of a negative log PMF of a discrete random variable, we can guarantee that $L_{\text{SINCEREideal}}(\theta) \geq 0$. However, we can prove a potentially tighter lower bound that depends on two quantities that define the difficulty of the contrastive learning task: the number of noise examples $|\mathcal{N}|$ and the symmeterized KL divergence between the two true data-generating distributions: the target distribution $p^+$ and the noise distribution $p^-$.

**Theorem 11.** *For any parameter $\theta$ of the tractable model, let $L_{SINCEREideal}(\theta)$ denote the ideal SINCERE loss in equation 10, computed via expectation over $\mathcal{X}, S, \mathcal{P}$ from the true model in equation 7. Then we have*

$$L_{SINCEREideal}(\theta) \geq \log |\mathcal{N}| - \big(KL(p^-||p^+) + KL(p^+||p^-)\big) \tag{11}$$

*where we recognize the sum of the two KL terms as the symmeterized KL divergence between $p^+$ and $p^-$, the true data-generating PDFs for individual images $x_i \in \mathcal{X}$.*

*Proof: See App. C.*

We emphasize that a proper bound is guaranteed throughout all steps in the proof. Previous bounds for InfoNCE loss by van den Oord et al. (2018) required some steps where the bound held only approximately. App. C provides further detail and interpretation.

The bound sensibly suggests that the minimizing loss value should increase as the number of noise samples $|\mathcal{N}|$ grows, because the model has a harder selection task due to choosing among more alternatives. If $p^+$ and $p^-$ are known, this bound indicates what loss values are achievable based on the separability of the distributions. If the right hand side of the bound evaluates to a negative number, a tighter bound is possible by invoking the fact that $L_{\text{SINCEREideal}}(\theta) \geq 0$. We caution that evaluations of $L_{\text{SINCEREideal}}(\theta)$ will be inexact in practice due to approximations of the ideal expectation.

This bound can also be used when $p^+$ and $p^-$ are unknown. As a corollary, a concrete numerical value of $\log |\mathcal{N}| - L_{\text{SINCEREideal}}(\theta)$ can be interpreted as a lower bound on the symmetrized KL between the unknown $p^+$ and $p^-$. Thus, a well-optimized model fit by minimizing SINCERE can provide a bound on a notion of divergence even when $p^+$ and $p^-$ are not available and difficult to estimate directly due to high-dimensionality of the data space. We find it interesting that a loss computed from the scalar output of a neural net trained to solve a selection task be used to bound the divergence between distributions over a much higher dimensional data space. Similar bounds have been reported for variational divergence minimization in generative-adversarial networks (Nowozin et al., 2016).

The previous mutual information bound for InfoNCE (van den Oord et al., 2018) has motivated further theoretic investigations of contrastive losses (Lee et al., 2024; Wu et al., 2020) and new contrastive learning methods (Yang et al., 2022a; Tian et al., 2020b; Murthy, 2021; Sordoni et al., 2021), particularly leading to applications with graph data (Xu et al., 2024), 3D data (Sanghi, 2020), and federated learning (Louizos et al., 2024). Our proposed KL divergence bound enables future work to utilize the relationship between target and noise distributions of supervised and self-supervised tasks in addition to the existing mutual information bound relating data samples and self-supervised instance discrimination labels.

### 3.3 SINCERE Loss in Practice

The above analysis assumes a single target distribution of interest. In practice, a supervised classification problem with $K$ classes requires learning $K$ target-noise separation tasks. Following Khosla et al. (2020), we assume one shared function $f_\theta$ approximates all $K$ target-noise tasks for simplicity.

To approximate the expectation needed for the SINCERE loss in practice, we average over stochastically-sampled mini-batches $(\mathcal{X}_b, \mathcal{Y}_b)$ of fixed size $N$ from a much larger labeled data set. These mini-batches are constructed from $N/2$ images that then have 2 randomly sampled data augmentations applied to them. This incorporates terms similar to self-supervised instance discrimination (Chen et al., 2020a) into SINCERE, as there will be $N$ terms where 2 augmentations of the same image form the target distribution samples.

Within each batch, we allow each index a turn as the selected target index $S$. For that turn, we define the target distribution as examples with class $y_S$ and the noise distribution as examples from other classes.

We thus fit neural net weights $\theta$ by minimizing this expected loss over batches:

$$\mathbb{E}_{\mathcal{X}_b, \mathcal{Y}_b} \left[ \sum_{S=1}^{N} \sum_{p \in \mathcal{P}} \frac{1}{N|\mathcal{P}|} L_{\text{SINCERE}}(z_S, z_p) \right], \tag{12}$$

$$L_{\text{SINCERE}}(z_S, z_p) = -\log \frac{e^{z_S \cdot z_p / \tau}}{e^{z_S \cdot z_p / \tau} + \sum_{n \in \mathcal{N}} e^{z_n \cdot z_p / \tau}}.$$

Our implementation of SINCERE uses the cosine similarity proposed by Wu et al. (2018) and averages over all same-class partners in $\mathcal{P}$. Other choices of similarity functions or pooling could be considered in future work. SINCERE and SupCon have the same complexity in both speed and memory, as detailed in App. E.

Averaging over the elements of $\mathcal{P}$ nonparametrically represents the target class $y_S$ and encourages each $z_p$ to have an embedding similar to its same-class partner $z_S$. However, no member of $\mathcal{P}$ ever appears in the denominator without also appearing in the numerator. This avoids any repulsion between two members of the same class in the embedding space seen with SupCon loss. Intuitively, our SINCERE loss restores NCE's assumption that the input used in the numerator belongs to the target distribution while all other inputs in the denominator belong to the noise distribution.

### 3.4 Analysis of Gradients

We study the gradients of both SINCERE and SupCon to gain additional understanding of their relative properties.

The gradient of the SINCERE loss with respect to $z_p$ is

$$\frac{z_S}{\tau}\left(\frac{e^{z_S \cdot z_p/\tau}}{\sum_{j \in \mathcal{N} \cup \{S\}} e^{z_j \cdot z_p/\tau}} - 1\right) + \frac{\sum_{n \in \mathcal{N}} \frac{z_n}{\tau} e^{z_n \cdot z_p/\tau}}{\sum_{j \in \mathcal{N} \cup \{S\}} e^{z_j \cdot z_p/\tau}}. \tag{13}$$

The first term has a *negative* scalar times $z_S$. The second term has a *positive* scalar times each noise embedding $z_n$. Thus each gradient descent update to $z_p$ encourages it to move *towards* the other target embedding $z_S$ and *away* from each noise embedding $z_n$. The magnitude of these movements is determined by the softmax of cosine similarities. For a complete derivation and further analysis, see App. D.

This behavior is different from the gradient dynamics of SupCon loss. Khosla et al. (2020) provide SupCon's gradient with respect to $z_p$ as

$$\frac{z_S}{\tau}\left(\frac{e^{z_S \cdot z_p/\tau}}{\sum_{i \in \mathcal{I}} e^{z_i \cdot z_p/\tau}} - \frac{1}{|\mathcal{P}|}\right) + \frac{\sum_{n \in \mathcal{N}} \frac{z_n}{\tau} e^{z_n \cdot z_p/\tau}}{\sum_{i \in \mathcal{I}} e^{z_i \cdot z_p/\tau}}. \tag{14}$$

The scalar multiplying $z_S$ in equation 14 will be in the range $[\frac{-1}{|\mathcal{P}|}, 1 - \frac{1}{|\mathcal{P}|}]$. The possibility of positive values implies intra-class repulsion: $z_p$ could be *pushed away* from $z_S$ when applying gradient descent. In contrast, the scalar multiplier for $z_S$ will always by in $[-1, 0]$ for SINCERE in equation 13, which effectively performs hard positive mining (Schroff et al., 2015). For further analysis of SupCon's gradient, see App. G.

### 3.5 Related Work

#### 3.5.1 Sampling Bias in Unsupervised Instance Discrimination

For self-supervised instance discrimination, Chuang et al. (2020) describe a problem they call "sampling bias," where several instances in the same batch treated as negative examples may, in reality, have the same unobserved supervised class label as the target instance. For example, in standard InfoNCE two distinct "dog" images will have their embeddings repelled from each other and drawn towards only their augmentation. Sampling bias induces the same effect on embeddings as the intra-class repulsion issue we raise in this work, but is driven by the fact that supervised classification labels are unknown, instead of improper formulation of a supervised loss when labels are known.

Interestingly, both our work and several others propose revised contrastive objectives that adjust a softmax denominator. Chuang et al. (2020) suggest a *debiased contrastive* objective that subtracts additional target distribution terms from InfoNCE's denominator to debias the noise distribution. In contrast, SINCERE removes extraneous target distribution terms from SupCon's denominator. Yeh et al. (2022) propose a *decoupled* objective that removes all target distribution terms from the denominator, seeking to improve learning efficiency without the need for large batches or many epochs. Future work could investigate if this idea works in a supervised context or has a probabilistic justification.

Arora et al. (2019) include sampling bias in their generative model for self-supervised contrastive learning via latent class random variables. Their contribution is to describe how a sufficiently large number of samples used in self-supervised contrastive learning can enable strong transfer learning performance in downstream supervised tasks, despite the bias. They require the assumption that the explicit class labels of the downstream task match the latent classes of the self-supervised task. In contrast, we model self-supervised and supervised contrastive learning as separate tasks to examine how the former can be generalized to latter.

#### 3.5.2 Supervised Contrastive Losses

Several works have expanded on SupCon loss in order to apply it to new problems. Feng et al. (2022) limit the target and noise distributions to K-nearest neighbors to allow for multi-modal class distributions. Kang

et al. (2021) explicitly set the number of samples from the target distribution to handle imbalanced data sets. Li et al. (2022b) introduce a regularization to push target distributions to center on uniformly distributed points in the embedding space. Yang et al. (2022b) and Li et al. (2022a) utilize pseudo-labeling to address semi-supervised learning and supervised learning with noisy labels respectively. SINCERE loss can easily replace the use of SupCon loss in these applications.

Terms similar to the SINCERE loss have previously been used as a part of more complex losses. Chen et al. (2022) utilizes a loss like our SINCERE loss as one term of an overall loss function meant to spread out embeddings that share a class. Detailed discussion or motivation for the changes made to SupCon loss is not provided. They do not identify or discuss the intra-class repulsion issue that motivates our work.

Barbano et al. (2023) proposed the $\epsilon$-SupInfoNCE loss to do supervised contrastive learning on datasets which are "biased" in the sense that some visual features spuriously correlate with class labels in available data but don't characterize the true data distribution. Motivated by metric learning, their loss seeks to ensure that a target-target pair of embeddings has cosine similarity at least $\epsilon$ larger than the closest target-noise pair, where $\epsilon > 0$ is their margin hyperparameter. Writing this goal as a maximum over target-noise pairs that is smoothly approximated by a LogSumExp function leads to their proposed loss, written in our notation as

$$L_{\epsilon\text{-SupInfoNCE}}(z_S, z_p) = -\log \frac{e^{z_S \cdot z_p/\tau}}{e^{z_S \cdot z_p/\tau - \epsilon} + \sum_{n \in \mathcal{N}} e^{z_n \cdot z_p/\tau}}. \tag{15}$$

Functionally, this loss is equivalent to our SINCERE loss when $\epsilon = 0$, though Barbano et al. advocate for larger $\epsilon$ and in fact do not try $\epsilon = 0$ for image classification in their Table 8.

Barbano et al. (2023) do not identify the intra-class repulsion issue that motivates our work and do not establish any probabilistic modeling foundations for their proposed loss. They justify removing SupCon's problematic target-target terms from the denominator only by calling these "non-contrastive." Ultimately, Barbano et al. (2023) focuses on using $\epsilon$-SupInfoNCE loss with an additional regularization loss for "debiasing," avoiding spurious correlations. Given this goal, their experiments do not investigate differences between $\epsilon$-SupInfoNCE loss and SupCon loss beyond supervised classification accuracy. Our transfer learning experiments in Sec. 4.2 find that hyperparameter search over non-zero $\epsilon$ values does not improve accuracy over SINCERE, where $\epsilon = 0$, on a majority of datasets.

## 4  Experiments

We compare our proposed SINCERE to two earlier supervised contrastive losses: SupCon (Khosla et al., 2020) and $\epsilon$-SupInfoNCE (Barbano et al., 2023). Sec. 4.1 shows how SINCERE separates the target and noise distributions in the learned embedding space better than SupCon. We do not compare against $\epsilon$-SupInfoNCE for this experiment as it does not use a softmax-based formulation and therefore does not have comparable similarity values. Sec. 4.2 evaluates transfer learning with a linear classifier, finding SINCERE outperforms SupCon on all 6 tested data sets and $\epsilon$-SupInfoNCE on 4 out of 6 tested data sets, even when $\epsilon$-SupInfoNCE is allowed to tune the value of its additional $\epsilon$ hyperparameter not present in SINCERE. Together, these two results support our main claims: that SINCERE repairs the intra-class repulsion issue of SupCon and offers representations that generalize well to new tasks. All tables bold results only when they are statistically significant based on the 95% confidence interval of the accuracy difference (Foody, 2009) from 1,000 iterations of test set bootstrapping.

Additional experiments on classifier *accuracy* are included in the appendix. Results there suggest SINCERE is just as good as SupCon and $\epsilon$-InfoNCE for standard supervised classification. We found no statistically significant difference in accuracy between SINCERE and the best other contrastive loss across 3 data sets, using both linear probing (App. L) and k-nearest neighbor classifier (App. M) evaluations. We argue the lack of difference in accuracy is unsurprising given the separations seen in Sec. 4.1: even though SINCERE learns greater separation between target and noise, SupCon separates target and noise well enough to get similar accuracy. In other experiments, App. K compares numerical values of SINCERE and SupCon losses after training, finding that intra-class repulsion raises SupCon's numerical loss value at the estimated minima.

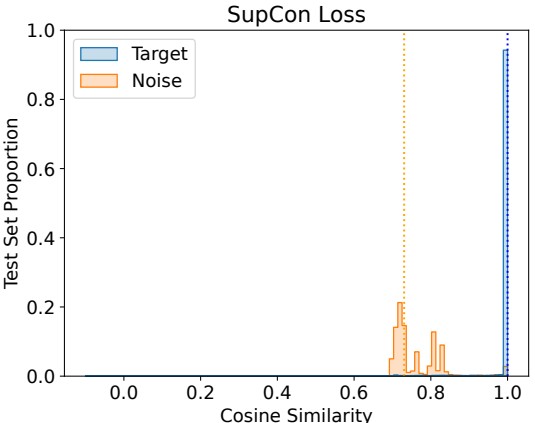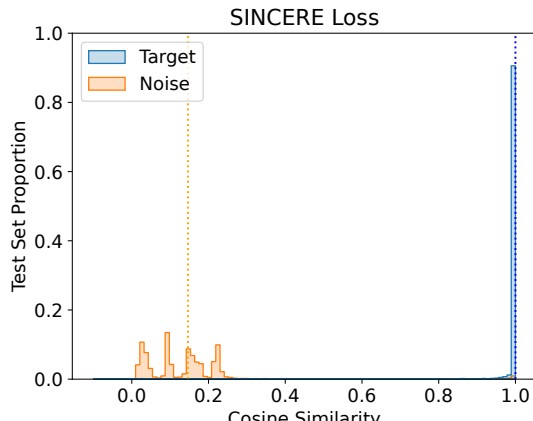

Figure 2: Histograms of cosine similarity values for CIFAR-10 test set nearest neighbors, comparing SupCon (left) and SINCERE (right). We plot the similarity of each test image to the nearest target image in the training set as well as the nearest noise image in the training set. The vertical dotted lines visualize the median similarity value. Our SINCERE loss reduces similarity to the nearest noise image by a substantial amount, thereby improving target-noise separation.

| Training Loss | ResNet | Batch Size | CIFAR-2 | CIFAR-10 | CIFAR-100 | ImageNet-100 |
|---------------|--------|-----------|---------|----------|-----------|--------------|
| SupCon | 50 | 512 | 0.410 | 0.270 | 0.438 | 0.026 |
| SINCERE | 50 | 512 | **0.972** | **0.854** | **0.454** | **0.154** |
| SupCon | 200 | 512 | - | 0.278 | 0.514 | 0.033 |
| SINCERE | 200 | 512 | - | **0.948** | 0.515 | **0.179** |
| SupCon | 50 | 1024 | - | 0.247 | 0.443 | 0.026 |
| SINCERE | 50 | 1024 | - | **0.921** | **0.515** | **0.166** |

Table 1: Separation between target and noise distributions (higher is better). Each number reports the average margin across target classes in the heldout test set. For each class, the margin is the difference between the median cosine similarity value of members of the target class and the corresponding median of members of the noise class; visually, the margin is the distance between vertical lines in Figure 2. SINCERE significantly increases the size of the margin on all tested data set, batch size, and architecture configurations except CIFAR-100 with ResNet-200. Cells marked "-" were not evaluated to limit expensive experiments.

For source datasets, our embedding models are trained on CIFAR-10, CIFAR-100 (Krizhevsky, 2009), and ImageNet-100 (Tian et al., 2020a; Chun-Hsiao Yeh, 2022) datasets, in all cases working with $32 \times 32$ pixel RGB images for expediency. A subset of CIFAR-10 containing only cat and dog images, referred to as CIFAR-2 here, was selected to evaluate binary classification. SupCon loss' problematic intra-class repulsion should be most pronounced on CIFAR-2 due to having the largest number of images sharing the same class.

Experiments primarily use ResNet-50 (He et al., 2016) architectures, with some results using bigger ResNet-200 models. Both were used in previous SupCon evaluations (Khosla et al., 2020). App. J provides details of the training process to aid in reproducing results with our shared code.

### 4.1 Target-Noise Separation in Learned Embedding Space

We examine the decision process for a 1 nearest-neighbor classifier in the learned embedding space to investigate how well each loss achieves target-noise separation. In Figure 2 we visualize how similarities of embedding pairs differ when using nearest neighbor representatives of both target and noise classes. For each

| Training Loss | Pet-37 | DTD-47 | Aircraft-100 | Food-101 | Flowers-102 | Cars-196 |
|---|---|---|---|---|---|---|
| SupCon | 53.91 | 50.73 | 38.60 | 62.91 | 64.96 | 46.14 |
| $\epsilon$-SupInfoNCE | **57.00** | **55.00** | 42.62 | 64.07 | 66.17 | 51.74 |
| SINCERE | 55.77 | 51.41 | **44.88** | **64.32** | **68.08** | **53.04** |

Table 2: Top-1 accuracy for transfer learning from ImageNet-100 using 32×32 resolution images. We use $\epsilon$=0.25 for $\epsilon$-SupInfoNCE, selected via hyperparameter search. Number of classes in each data set appended if not part of the data set's name. Bolded results are statistically significant based on the bootstrapped 95% confidence interval of the accuracy difference. The original SupCon paper (Khosla et al., 2020) reports results for larger image resolutions, infeasible without an industrial GPU, and thus their numbers are not directly comparable.

CIFAR-10 test set image, we plot similarity to its nearest neighbor from the train set target and train set noise distributions, labeled as "Target" and "Noise" respectively.

Examining Fig. 2, we find that both SINCERE and SupCon losses succeed at maximizing the cosine similarity of target-target pairs, with histograms very close to the max value of 1. However, SINCERE noticeably lowers the cosine similarity of target-noise pairs, with values in the 0-0.25 range instead of SupCon's 0.7-0.85 range. This visualization shows that SINCERE achieves a wider margin of target-noise separation on average across all 10 classes. App. O provides visualizations of class-specific histograms; we find each class' behavior mimics the aggregate findings here of wider target-noise separation for SINCERE.

Table 1 quantifies target-noise separation by measuring the margin between the median similarity values of the target and noise distribution. SINCERE loss leads to a significantly larger margin for all combinations of data sets, batch sizes, and architectures tested except CIFAR-100 with ResNet-200, where there is no significant difference. This shows SINCERE is effective across architectures and batch sizes. This confirms our intuitive picture in Fig. 1 and the analysis of gradients in Sec. 3.4: SupCon's problematic inclusion of some target images as part of the noise distribution reduces target-noise separation compared to SINCERE.

SupCon and SINCERE both have a positive margin between target and noise nearest neighbors, leading to both methods creating accurate classifiers on all tested datasets, as shown in App. L and App. M. Intuitively, even though SINCERE desirably improves target-noise separation, it does not produce notably different classification decisions on the tested data sets; SupCon appears to separate enough to get decent accuracy.

## 4.2 Transfer Learning

Transfer learning aims to increase performance on a target task by utilizing knowledge from a different source task (Zhuang et al., 2021). We evaluate how supervised training on a source task with different contrastive losses impacts target task performance. We utilize linear probing, which trains a linear classifier with the frozen embedding function from the source task model, to evaluate how well each loss's learned embedding function generalizes to new target tasks.

We choose classification on ImageNet-100 (Tian et al., 2020a; Chun-Hsiao Yeh, 2022) as our source task to evaluate transfer learning with a large source data set. Images are resized to 32 by 32 pixels to accommodate a 512 batch size without exceeding GPU memory constraints. Some previous works (Chen et al., 2020a; 2021; Khosla et al., 2020) utilize much larger 224 by 224 pixel images for training on large GPU clusters at an industrial-scale research company. This resolution difference makes our results not directly comparable. Fully training a model on ImageNet-100 even at $32 \times 32$ took one week of computation on the hardware available at our academic institution, outlined in App. J.

Table 2 reports the accuracy for linear probing on various target data sets with ImageNet-100 as the source data set. SINCERE outperforms SupCon on every data set tested, suggesting that greater target-noise separation on the source task enables better linear classification in the same embedding space for target tasks.

We also compare against $\epsilon$-SupInfoNCE (Barbano et al., 2023), which is more flexible than SINCERE due to the additional $\epsilon$ margin hyperparameter. $\epsilon$-SupInfoNCE reduces to SINCERE loss when $\epsilon = 0$, but this

setting for $\epsilon$ was not evaluated in previous work. We limited the hyperparameter search for $\epsilon$ to $[0.1, 0.25, 0.5]$, as was done in Barbano et al. (2023), to avoid similarity in losses as $\epsilon$ approaches 0.

SINCERE improves accuracy on 4 out of the 6 target data sets tested: FGVC-Aircraft (Maji et al., 2013), Food-101 (Bossard et al., 2014), Flowers-102 (Nilsback & Zisserman, 2008), and Cars (Krause et al., 2013). On two other data sets, Pets (Parkhi et al., 2012) and the Describable Textures Dataset (Cimpoi et al., 2014), $\epsilon$-SupInfoNCE performs best. This shows an additional hyperparameter search for values of $\epsilon > 0$ can enable $\epsilon$-SupInfoNCE to outperform SINCERE on some data sets. However, the better overall performance of SINCERE suggests that expensive hyperparameter tuning is often unnecessary.

## 5   Discussion

The proposed SINCERE loss is a theoretically motivated loss for supervised noise contrastive estimation. Compared to the previous SupCon loss for the same task, SINCERE eliminates problematic repulsion of examples that share a class label while delivering better target-noise separation. For practitioners, SINCERE loss can be a drop-in replacement for SupCon loss. There are no apparent downsides to switching to SINCERE: runtime and memory complexity will be indistinguishable, as will source-task classifier accuracy, while transfer learning is likely to improve. We do suggest carefully refitting loss-weight hyperparameters for multi-term losses due to SINCERE's broader range of values.

Separating classes in embedding space is the stated goal of supervised contrastive learning. From the abstract of the SupCon paper by Khosla et al., the goal is to create embeddings where "Clusters of points belonging to the same class are pulled together in embedding space, while simultaneously pushing apart clusters of samples from different classes." We argue that while our work and SupCon share this goal of target-noise separation, SupCon loss does not achieve it as well as our SINCERE loss, as evidenced by the histogram plots in our Fig. 2 and the quantitative separation values in Tab. 1.

Favoring the widest possible separation of classes in the embedding space is preferred based on the *maximum margin* principle that has been a foundation of classifier representation learning for decades (Boser et al., 1992; Cortes, 1995; Smola, 2000). There are intuitive arguments for favoring large margins (see Sec 17.3.1 of Murphy's Probabilistic Machine Learning textbook). There are also more formal arguments related to guarantees about how well a classifier might do on test data. Many works like Wei & Ma (2019) have proven generalization bounds that scale inversely polynomial with the margin size (Glasgow et al., 2022). In other words: these bounds suggest that wider margins between classes can imply lower classification error on never-before-seen data drawn from the same distribution as the train set. We hope our work might inspire future work on such generalization bounds for supervised contrastive learning.

We acknowledge several limitations in the scope of this work due to time and budget constraints. Our experiments focus on supervised contrastive methods for image classification with a ResNet (He et al., 2016) architecture. Architectures such as the vision transformer (ViT) (Dosovitskiy et al., 2021) may slightly improve performance, but are more difficult to successfully train (Chen et al., 2021). We do not evaluate on other data modalities such as text (Gunel et al., 2021; Jiang et al., 2021) or graph (You et al., 2020; 2021; Xu et al., 2021) data, although InfoNCE and SupCon have been used successfully in those domains. Due to our focus on representation learning, we do not compare against methods such as cross-entropy loss that do supervised classification without explicitly manipulating the embedding space.

Future work may explore an alternative supervised loss which predicts all members of the target distribution at once instead of individually. A naive approach to this problem would involve an exponential increase in the number of terms in the denominator, but could potentially model higher-order interactions between sets of samples instead of averaging over pair-wise interactions as is done currently. Other works may examine intentionally introducing a repulsion between target examples, such as jointly modeling instance discrimination and supervised contrastive similarities. Further investigation is also possible for how self-supervised methods with similar loss structures, such as BYOL (Grill et al., 2020) and Decoupled Contrastive Learning (Yeh et al., 2022), could be derived by defining self-supervised probabilistic models similar to ours.

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

## Appendix Contents

| Notation | Definition |
|---|---|
| $N$ | number of elements in the data set |
| $K$ | number of labels such that $2 \leq K \leq N$ |
| $(\mathcal{X}, \mathcal{Y})$ | observed data set with $N$ elements |
| $\mathcal{X} = (x_1, x_2, ..., x_N)$ | data (e.g. images) |
| $\mathcal{Y} = (y_1, y_2, ..., y_N)$ | categorical labels in $[\![1, K]\!]$ |
| $\mathcal{I} = [\![1, N]\!]$ | set of indices for elements in the data set |
| $D$ | dimensionality of neural network embeddings |
| $z_i \in \mathbb{R}^D$ | neural network unit vector embedding of $x_i$ |
| $\mathcal{T}$ | indices for target distribution samples |
| $\mathcal{N} = \mathcal{I} \setminus \mathcal{T}$ | indices for noise distribution samples |
| $T$ | defines the number of samples from the target distribution such that $2 \leq T \leq N - 1$ |
| $\mathcal{P} \in \{I \subset \mathcal{I} \mid |I| = T - 1\}$ | random variable for the set of indices for same-class partners for $S$ |
| $S \in \mathcal{I} \setminus \mathcal{P}$ | random variable defining the index of the target sample of interest |
| $f(x_i, y_j)$ | score function outputting a scalar score representing how well data $x_i$ matches class representation $y_j$ |
| $\tau$ | temperature hyperparameter, typically about 0.1 in practice |
| $i \in \mathcal{I}$ | index for arbitrary element of the data set |
| $p \in \mathcal{P}$ | index from the partners for $S$ |
| $n \in \mathcal{N}$ | index from the noise distribution |
| $j$ | index used for clarity when another index notation already used, such as nested summations |
| $p^+(x_i)$ | target data likelihood |
| $p^-(x_i)$ | noise data likelihood |
| $p(\mathcal{X}|S)$ | data generating model for the self-supervised case |
| $p(S|\mathcal{X})$ | likelihood index $S$ is the index of the target class sample for the self-supervised case |
| $p(\mathcal{X}|\mathcal{P}, S)$ | data generating model for the supervised case |
| $p(S|\mathcal{X}, \mathcal{P})$ | likelihood $S$ is the index of the last target class sample for the supervised case |
| $\theta$ | parameters of neural network $f_\theta$ |
| $f_\theta(x_i, y_S)$ | neural network used in the tractable model of the index likelihoods |

Table 3: Notation reference with abridged definitions.

## A    Notation Reference

See Table 3.

# B  Proof of Proposition 6

Given the assumed model in equation 7, we wish to show that the probability that a specific index $S$ is the last remaining index of the target class is

$$p(S|\mathcal{X}, \mathcal{P}) = \frac{\frac{p^+(x_S)}{p^-(x_S)}}{\frac{p^+(x_S)}{p^-(x_S)} + \sum_{n \in \mathcal{N}} \frac{p^+(x_n)}{p^-(x_n)}} \tag{16}$$

where the set of "negative" indices is defined as $\mathcal{N} = \{1, 2, \ldots N\} \setminus (\mathcal{P} \bigcup \{S\})$. We emphasize that $\mathcal{N}$ is determined by $\mathcal{P}$, the set of positive indices (other examples of the target class).

**Proof** We can define the joint over $S$ and $\mathcal{P}$ given data set $\mathcal{X}$ via Bayes' rule manipulations

$$p(S, \mathcal{P}|\mathcal{X}) = \frac{p(\mathcal{X}, S, \mathcal{P})}{p(\mathcal{X})} = \frac{p(\mathcal{X}|S, \mathcal{P})p(S, \mathcal{P})}{p(\mathcal{X})} \tag{17}$$

Plugging in basic model definitions from equation 7 into the numerator, and using shorthand $u > 0$ to represent the uniform probability mass produced by evaluating $p(S, \mathcal{P})$ at any valid inputs, we have

$$p(S, \mathcal{P}|\mathcal{X}) = \frac{\prod_{i \in \mathcal{P} \bigcup \{S\}} p^+(x_i) \prod_{n \in \mathcal{N}} p^-(x_n) \cdot u}{\sum_{\mathcal{R} \in \mathbb{P}_T} \left( \prod_{r \in \mathcal{R}} p^+(x_r) \prod_{m \in \mathcal{I} \setminus \mathcal{R}} p^-(x_m) \cdot u \right)} \tag{18}$$

where $\mathbb{P}_T$ denotes the set of all possible subsets of indices $\mathcal{I}$ with size exactly equal to $T$.

Next, apply two algebraic simplifications. First, cancel the $u$ terms from both numerator and denominator. Second, multiply both numerator and denominator by $\prod_{a \in \mathcal{I}} \frac{1}{p^-(x_a)}$ (a legal move with net effect of multiply by 1). After grouping each product into terms with ratio of $p^+/p^-$ (which remain) and terms with $p^-/p^-$ (which cancel away), we have

$$p(S, \mathcal{P}|\mathcal{X}) = \frac{\prod_{i \in \mathcal{P} \bigcup \{S\}} \frac{p^+(x_i)}{p^-(x_i)}}{\underbrace{\sum_{\mathcal{R} \in \mathbb{P}_T} \left( \prod_{r \in \mathcal{R}} \frac{p^+(x_r)}{p^-(x_r)} \right)}_{\Omega}} = \frac{1}{\Omega} \prod_{i \in \mathcal{P} \bigcup \{S\}} \frac{p^+(x_i)}{p^-(x_i)} \tag{19}$$

For convenience later, we define the denominator of the right hand side as $\Omega$, which is a constant with respect to $S$ and $\mathcal{P}$.

Now, we wish to pursue our goal conditional of interest: $p(S|\mathcal{P}, \mathcal{X})$. Using Bayes rule on the joint $p(S, \mathcal{P}|\mathcal{X})$ above, we have

$$p(S|\mathcal{P}, \mathcal{X}) = \frac{p(S, \mathcal{P}|\mathcal{X})}{p(\mathcal{P}|\mathcal{X})} \tag{20}$$

$$= \frac{p(S, \mathcal{P}|\mathcal{X})}{\sum_{j \in \mathcal{I} \setminus \mathcal{P}} p(S = j, \mathcal{P}|\mathcal{X})} \tag{21}$$

$$= \frac{\frac{1}{\Omega} \prod_{i \in \mathcal{P} \bigcup \{S\}} \frac{p^+(x_i)}{p^-(x_i)}}{\frac{1}{\Omega} \sum_{j \in \mathcal{I} \setminus \mathcal{P}} \prod_{\ell \in \mathcal{P} \bigcup \{j\}} \frac{p^+(x_\ell)}{p^-(x_\ell)}} \tag{22}$$

Finally, canceling terms that appear in both numerator and denominator (the $\Omega$ term as well as the product over $\mathcal{P}$), this leaves

$$p(S|\mathcal{P}, \mathcal{X}) = \frac{\frac{p^+(x_S)}{p^-(x_S)}}{\sum_{j \in \mathcal{I} \setminus \mathcal{P}} \frac{p^+(x_j)}{p^-(x_j)}} = \frac{\frac{p^+(x_S)}{p^-(x_S)}}{\frac{p^+(x_S)}{p^-(x_S)} + \sum_{n \in \mathcal{N}} \frac{p^+(x_n)}{p^-(x_n)}} \tag{23}$$

Where the last statement follows because $\mathcal{I} \setminus \mathcal{P} = S \bigcup \mathcal{N}$ by definition of $\mathcal{N}$. We have thus reached the desired statement of equality. ∎

## C  Proof of Bound Relating SINCERE to Negative KL in Theorem 11

Assume the target class is known and fixed throughout this derivation. Further assume that both the target and noise distribution provide support over all possible data inputs $x$, so $p^+(x) > 0$ and $p^-(x) > 0$.

We start with the definition of the loss as an expected negative log likelihood of the selected index $S$ from equation 10.

$$L(\theta) = \mathbb{E}_{\mathcal{X},S,\mathcal{P} \sim p_{true}}[-\log p_\theta(S|\mathcal{P},\mathcal{X})] \tag{24}$$

where the expectation is with respect to samples $\mathcal{X}, S, \mathcal{P}$ from the joint of the "true" model defined in equation 7. We denote the loss as $L(\theta)$ in this section to emphasize that the proof applies to both SINCERE and InfoNCE losses.

Recall the optimal tractable model with weights $\theta^*$ defined via target-to-noise density ratios in Prop. 9. The loss at this parameter is a lower bound of the loss at any parameter: $L(\theta) \geq L(\theta^*)$.

Now, substituting the definition of $\theta^*$, we find that by simplifying via algebra

$$L(\theta) \geq L(\theta^*) = \mathbb{E}_{\mathcal{X},S,\mathcal{P}}[-\log p_{\theta^*}(S|\mathcal{P},\mathcal{X})] \tag{25}$$

$$= \mathbb{E}_{\mathcal{X},S,\mathcal{P}}\left[-\log \frac{\frac{p^+(x_S)}{p^-(x_S)}}{\frac{p^+(x_S)}{p^-(x_S)} + \sum_{n \in \mathcal{N}} \frac{p^+(x_n)}{p^-(x_n)}}\right]$$

$$= \mathbb{E}_{\mathcal{X},S,\mathcal{P}}\left[-\log \frac{1}{1 + \frac{1}{\frac{p^+(x_S)}{p^-(x_S)}} \sum_{n \in \mathcal{N}} \frac{p^+(x_n)}{p^-(x_n)}}\right]$$

$$= \mathbb{E}_{\mathcal{X},S,\mathcal{P}}\left[\log\left(1 + \frac{p^-(x_S)}{p^+(x_S)} \sum_{n \in \mathcal{N}} \frac{p^+(x_n)}{p^-(x_n)}\right)\right]$$

Next, we invoke another bound, using the fact that $\log 1 + p \geq \log p$ for any $p > 0$ (log is a monotonic increasing function).

$$L(\theta^*) \geq \mathbb{E}_{\mathcal{X},S,\mathcal{P}}\left[\log\left(\frac{p^-(x_S)}{p^+(x_S)} \sum_{n \in \mathcal{N}} \frac{p^+(x_n)}{p^-(x_n)}\right)\right] \tag{26}$$

$$= \underbrace{\mathbb{E}_{\mathcal{X},S,\mathcal{P}}\left[\log \sum_{n \in \mathcal{N}} \frac{p^+(x_n)}{p^-(x_n)}\right]}_{A} + \underbrace{\mathbb{E}_{\mathcal{X},S,\mathcal{P}}\left[\log \frac{p^-(x_S)}{p^+(x_S)}\right]}_{B}$$

We handle terms A then B separately below.

**Term A:**  Term A can be attacked by a useful identity: for any non-empty set of values $a_1, \ldots a_L$, such that all are strictly positive ($a_\ell > 0$), we can bound of log-of-sum as

$$\log\left(\sum_{\ell=1}^{L} a_\ell\right) \geq \log L + \frac{1}{L} \sum_{\ell=1}^{L} \log a_\ell \tag{27}$$

This identity is easily proven via Jensen's inequality (credit to user Kavi Rama Murthy's post on Mathematics Stack Exchange (Murthy, 2021)).

Using the above identity, our Term A of interest becomes

$$\text{term A} = \mathbb{E}_{\mathcal{X},S,\mathcal{P}}\left[\log\left(\sum_{n \in \mathcal{N}} \frac{p^+(x_n)}{p^-(x_n)}\right)\right] \tag{28}$$

$$\geq \mathbb{E}_{S,\mathcal{P}}\mathbb{E}_{\mathcal{X} \sim p(\mathcal{X}|S,\mathcal{P})}\left[\log |\mathcal{N}| + \frac{1}{|\mathcal{N}|} \sum_{n \in \mathcal{N}} \log \frac{p^+(x_n)}{p^-(x_n)}\right]$$

Under our model assumptions, the size of $\mathcal{N}$ is fixed to $N-T$ under Assumption 4 and does not fluctuate with $S$ or $\mathcal{P}$. Furthermore, recall that given any known value of the target index $S$, all data vectors corresponding to noise indices $\mathcal{N}$ are generated as i.i.d. draws from the noise distribution: $x_n \sim p^-(\cdot)$. These two facts plus linearity of expectations let us simplify the above as

$$\text{term A} \geq \log |\mathcal{N}| + \frac{1}{|\mathcal{N}|} \mathbb{E}_{S,\mathcal{P}} \left( \sum_{n \in \mathcal{N}} \int_{x_n} p^-(x_n) \left[ \log \frac{p^+(x_n)}{p^-(x_n)} \right] dx_n \right) \tag{29}$$

Next, realize that the inner integral is constant with respect to the indices choices defined by random variables $S, \mathcal{P}$. Furthermore, the sum over $n$ simply repeats the same expectation $|\mathcal{N}|$ times (canceling out the $\frac{1}{|\mathcal{N}|}$ term). This leaves a compact expression for a lower bound on term A:

$$\text{term A} \geq \log |\mathcal{N}| + \int_x p^-(x) \left[ \log \frac{p^+(x)}{p^-(x)} \right] dx \tag{30}$$
$$= \log |\mathcal{N}| - \text{KL}(p^-(x) || p^+(x)).$$

This reveals an interpretation of term A as a negative KL divergence from noise to target, plus the log of the size of negative set (a problem-specific constant).

**Term B:** The right-hand term B only involves the feature vector $x_S$ at the target index $S$ and not any other terms in $\mathcal{X}$. Thus, we can simplify the expectation over $p(\mathcal{X}|S, \mathcal{P})$ as follows

$$\text{term B} = \mathbb{E}_{\mathcal{X}, S, \mathcal{P}} \left[ \log \frac{p^-(x_S)}{p^+(x_S)} \right] = \mathbb{E}_{S,\mathcal{P}} \left[ \int_{\mathcal{X}} \log(\frac{p^-(x_S)}{p^+(x_S)}) p^+(x_S) \prod_{p \in \mathcal{P}} p^+(x_p) \prod_{n \in \mathcal{N}} p^-(x_n) d\mathcal{X} \right] \tag{31}$$
$$= \mathbb{E}_{S,\mathcal{P}} \left[ \int \log \left[ \frac{p^-(x_S)}{p^+(x_S)} \right] p^+(x_S) dx_S \right]$$

where we got all other $x_p$ and $x_n$ terms to simplify away because integrals over their PDFs evaluate to 1.

Now, we can recognize what remains above as a negative KL divergence. Let's write this out explicitly, replacing $x_S$ with notation $x$ (no subscript) simply to reinforce that the KL term inside the expectation does not vary with $S$ (regardless of which index is chosen, the target and noise distributions compared by the KL will be the same). This yields

$$\text{term B} = \mathbb{E}_{S,\mathcal{P}} \left[ \int \log \left[ \frac{p^-(x)}{p^+(x)} \right] p^+(x) dx \right] \tag{32}$$
$$= -\mathbb{E}_{S,\mathcal{P}}[\text{KL}(p^+(x) || p^-(x))] \tag{33}$$
$$= -\text{KL}(p^+(x) || p^-(x)) \tag{34}$$

where in our ultimate expression, we know the KL is constant w.r.t. $\mathcal{S}, \mathcal{P}$. Thus, the expectation simplifies away (writing out the full expectation as a sum then bringing the KL term outside leaves a PMF that sums to one over the sample space).

This reveals an interpretation of term B as a negative KL divergence from target to noise.

**Combining terms A and B.** Putting it all together, we find

$$L(\theta) \geq L(\theta^*) \geq \log |\mathcal{N}| - \underbrace{\left( \text{KL}(p^-(x) || p^+(x)) + \text{KL}(p^+(x) || p^-(x)) \right)}_{\text{symmeterized KL divergence}} \tag{35}$$

Thus, every evaluation of our proposed SINCERE loss has an information-theoretic interpretation as an upper-bound on the sum of the log of the size of the noise samples and the negative *symmeterized* KL divergence between the target and noise distributions. For a definition of symmeterized KL divergence see this link to Wikipedia

**Interpretation.** The bound above helps quantify what loss values are possible, based on two problem-specific elementary facts: the symmeterized divergence between target and noise distributions (where larger values mean the target-noise distinction is easier) and the total number of noise samples.

Naturally, the more trivial lower bound for $L(\theta)$ is zero, as that is the lowest any negative log PMF over any discrete variable (like $S$) can go. We observe that our proposed bound can often provide more information than this trival one, as large $|\mathcal{N}|$ will push the bound well above zero.

We further observe the following:

- Larger symmeterized KL will lower the RHS of the bound, indicating loss values can go lower. This intuitively makes sense: when target and noise distributions are easier to separate, we can get closer and closer to "perfect" predictions of $S$ with our tractable model and thus PMF values $p_\theta(S|\mathcal{X}, \mathcal{P})$ approach 1 and $L$ approaches 0.

- As the total number of noise samples gets higher, the RHS of the bound gets larger. This also makes sense, as the problem becomes harder (more chances to guess wrong when distinguishing between the one target sample and many noise samples), our expected loss should also increase.

**Relation to previous bounds derived for InfoNCE.** van den Oord et al. (2018) derive a bound relating their InfoNCE loss to a mutual information quantity in the self-supervised case. Indeed, our derivation of our bound was inspired by their work. Here, we highlight three key differences between our bound and theirs.

First, our bound applies to the more general supervised case, not just the self-supervised case.

Second, following their derivation carefully, notice that the claimed bound requires an *approximation* in their Eq. 8 in the appendix "A.1 Estimating the Mutual Information with InfoNCE" of van den Oord et al. (2018). While they argue this approximation becomes more accurate as batch size $N$ increases, indeed for any finite $N$ the claim of a strict bound is not guaranteed. In contrast, our entire derivation above requires no approximation.

Finally, the relation derived in van den Oord et al. (2018) is expressed in terms of mutual information, not symmeterized KL divergence. This is due to their choice to write the noise distribution as $p(x_i)$ and the target distribution as $p(x_i|c)$ where index $c$ denotes the target class of interest. For us, these two choices imply the noise and target are related by the sum rule

$$p(x_i) = \sum_{c'} p(c')p(x_i|c') \tag{36}$$

over an extra random variable $c$ whose sample space and PMF are not extremely clear, at least in our reading of van den Oord et al. (2018). In contrast, our formulation throughout Sec. 3.1 of the main paper fixes one target and one noise distribution throughout. We think this is a conceptually cleaner approach.

## D    Further Analysis of Gradients

The gradient of the SINCERE loss for a single $z_S, z_p$ pair with respect to $z_p$ is $\frac{\delta}{\delta z_S} L_{\text{SINCERE}}(z_S, z_p)$

$$= -\frac{\delta}{\delta z_p} \log \frac{e^{z_S \cdot z_p/\tau}}{e^{z_S \cdot z_p/\tau} + \sum_{i \in \mathcal{N}} e^{z_i \cdot z_p/\tau}} \tag{37}$$

$$= -\frac{\delta}{\delta z_p} \left( \frac{z_S \cdot z_p}{\tau} - \log \sum_{j \in \mathcal{N} \cup \{S\}} e^{z_j \cdot z_p/\tau} \right) \tag{38}$$

$$= \frac{-1}{\tau} \left( z_S - \frac{\sum_{j \in \mathcal{N} \cup \{S\}} z_j e^{z_j \cdot z_p/\tau}}{\sum_{j \in \mathcal{N} \cup \{S\}} e^{z_j \cdot z_p/\tau}} \right) \tag{39}$$

$$= \frac{-1}{\tau} \left( z_S - \frac{z_S \cdot e^{z_S \cdot z_p/\tau} + \sum_{i \in \mathcal{N}} z_i \cdot e^{z_i \cdot z_p/\tau}}{\sum_{j \in \mathcal{N} \cup \{S\}} e^{z_j \cdot z_p/\tau}} \right) \tag{40}$$

$$= \frac{1}{\tau} \left( z_S \left( \frac{e^{z_S \cdot z_p/\tau}}{\sum_{j \in \mathcal{N} \cup \{S\}} e^{z_j \cdot z_p/\tau}} - 1 \right) + \sum_{i \in \mathcal{N}} z_i \left( \frac{e^{z_i \cdot z_p/\tau}}{\sum_{j \in \mathcal{N} \cup \{S\}} e^{z_j \cdot z_p/\tau}} \right) \right). \tag{41}$$

We clarify the steps taken in this derivation. The first line simply substitutes the definition of $L_{\text{SINCERE}}(z_S, z_p)$. The second line breaks the fraction inside the log into two log terms, canceling with the exponentiation in the numerator term. The third line calculates the derivative with respect to $z_p$. The fourth line breaks up the sum in the numerator to highlight that there are two terms with $z_S$ as a vector ($z_S$ not in a dot product). The final line both brings the $-1$ inside the parentheses and combines the two terms with $z_S$ as a vector to clarify the weight pulling $z_p$ towards $z_S$.

## E    Runtime and Memory Complexity

Given a batch of $N$ data points, each with a $D$-dimensional embedding, SINCERE or SupCon loss can be computed in $O(N^2 D)$ time, with quadratic complexity arising due to need for computation of dot products between many pairs of embeddings. An implementation that was memory sensitive could be done with $O(ND)$ memory, which is the cost of storing all embedding vectors. Our implementation has memory cost of $O(N^2 + ND)$, as we find computing all $N^2$ pairwise similarities at once has speed advantages due to vectorization.

We more closely examine the change in the denominator calculation to show that it does not change the big-O complexity of SINCERE relative to SupCon. We perform this calculation after the computation of dot products between all pairs of embeddings described previously. SINCERE requires a unique denominator for each pair $S$ and $p$, so there are at most $N^2$ denominators to be calculated. For a given index $S$, the noise embeddings in the denominator will not change regardless of the choice of $p$. Therefore these at most $N$ noise terms are aggregated for each $S$ to create a "base" denominator, producing $N$ base denominators from $N$ operations each for $O(N^2)$ time. These $N$ base denominators then need to be combined with the unique $S \cdot P$ terms to complete the unique denominators used by SINCERE. There are at most $N^2$ unique denominators and each is calculated from a constant time operation on a $S \cdot p$ term and a base denominator, so $O(N^2)$ time is required. The calculation of the denominator terms is therefore dwarfed by the $O(N^2 D)$ time required by both methods to compute the pairwise dot products.

In our experiments with a batch size of 512, we find the runtime of computing embeddings with the forward pass of a neural network far exceeds the runtime of computing losses given embeddings.

## F Probabilistic View of SupCon

Attempting to translate SupCon loss into the noise-contrastive paradigm suggests that it assigns probability to the data point at index $S$ out of all possible data points via

$$\frac{\frac{p^+(x_S)}{p^-(x_S)}}{\frac{p^+(x_S)}{p^-(x_S)} + \sum_{j\in\mathcal{P}\setminus\{p\}}\frac{p^+(x_j)}{p^-(x_j)} + \sum_{n\in\mathcal{N}}\frac{p^+(x_n)}{p^-(x_n)}}. \tag{42}$$

We emphasize that this does *not* correspond to a principled derivation from a coherent probabilistic model. In fact, it is not a softmax over values of $S$ because the sum of these values over all valid $S$ is less than 1. In contrast, our derivation of SINCERE follows directly from the model in equation 7. Furthermore, this framing of SupCon makes clear that the additional denominator terms penalize similarity between embeddings from the target distribution, which results in the problematic intra-class repulsion behavior described in Fig. 1.

## G SupCon Gradient Analysis

SupCon's possible repulsion between members of the same class increases in severity as $|\mathcal{P}|$ increases, resulting in a scalar in $[0, 1]$ as $|\mathcal{P}|$ approaches positive infinity. Khosla et al. (2020) previously hypothesized that the $\frac{-1}{|\mathcal{P}|}$ term came from taking the mean of the embeddings $z_p \in \mathcal{P}$ . Our analysis suggests it is actually due to improperly including target class examples other than $S$ and $p$ in the loss' denominator.

A similar issue arises from the summation over the noise distribution in equation 14. Each softmax includes the noise distribution and the entire target distribution in the denominator instead of only the noise distribution and $z_p$ as in equation 13. This reduces the SupCon loss' penalty on poor separation between the noise and target distributions.

## H SupCon Bound Looser than SINCERE Bound

Applying the strategy from Sec. C to SupCon loss results in the following bound:

$$L^{\text{SupCon}} \geq \log(|\mathcal{N}| + |\mathcal{P}| - 1) - \frac{|\mathcal{N}|}{|\mathcal{N}| + |\mathcal{P}| - 1}\left(KL(p^-||p^+) + KL(p^+||p^-)\right). \tag{43}$$

This bound is greater than or equal to the SINCERE bound with equality only in the case of $\mathcal{P} = 1$, where the losses are equivalent.

## I SupCon Equation Notation

Khosla et al. (2020) write the SupCon loss in their notation as:

$$\sum_{i\in I}\frac{-1}{|P(i)|}\sum_{p\in P(i)}\log\frac{e^{z_i\cdot z_p/\tau}}{\sum_{a\in A(i)}e^{z_i\cdot z_a/\tau}}. \tag{44}$$

whereas the SupCon loss in our notation is:

$$\sum_{S=1}^{N}\frac{-1}{|\mathcal{P}|}\sum_{p\in\mathcal{P}}\log\frac{e^{z_S\cdot z_p/\tau}}{\left(\sum_{j\in\mathcal{T}\setminus\{p\}}e^{z_j\cdot z_p/\tau}\right) + \sum_{n\in\mathcal{N}}e^{z_n\cdot z_p/\tau}}. \tag{45}$$

The sets in each definition are equivalent: $I = [\![1, N]\!]$, $P(i) = \mathcal{P}$, and $A(i) = (\mathcal{T}\setminus\{p\})\cup\mathcal{N}$. The only remaining difference is the former chooses to use the outer sum variable $i$ in each term of the softmax while we choose the inner sum variable $p$. Their softmax given $i = \alpha$, $p = \beta$ is equivalent to our softmax given $S = \beta$, $p = \alpha$ for all indices $\alpha$, $\beta$ sharing a class. Since each pair $\alpha$, $\beta$ will appear in exactly one term of each summation, the definitions are equivalent.

| Training Loss | CIFAR-2 | | CIFAR-10 | | CIFAR-100 | | ImageNet-100 | |
|---|---|---|---|---|---|---|---|---|
| | Initial | Final | Initial | Final | Initial | Final | Initial | Final |
| SupCon | 6.94 | 6.28 | 6.94 | 4.69 | 6.91 | 2.37 | 6.92 | 2.35 |
| SINCERE | 6.25 | **0.99** | 6.80 | **0.29** | 6.91 | **0.10** | 6.91 | **0.11** |

Table 4: Average training loss values for initial and final training epochs. The final SINCERE loss value consistently approaches 0 regardless of the number of classes in the data set. Intra-class repulsion causes SupCon loss' minimum to increase with fewer classes, which is seen in practice in the large variation in final loss value.

## J   Training and Hyperparameter Selection

Models were trained on a Red Hat Enterprise Linux 7.5 server with a A100 GPU with 40 GiB of memory and 16 Intel Xeon Gold 6226R CPUs. Many of the CPUs were primarily used for parallelization of data loading, so fewer or smaller CPUs could be used easily. PyTorch 2.0.1 and Torchvision 0.15.2 for CUDA 12.1 were used for model and loss implementations.

A hyperparameter search was done for each loss with 10% of the training set used as validation. Training was done with 800 epochs of stochastic gradient descent with 0.9 momentum, 0.0001 weight decay, and a cosine annealed learning rate schedule with warm-up, which spends 10 epochs warming up from 0.1% to 100% then cosine anneals back to 0.1% at the last epoch. Unless marked otherwise, all experiments used batch size 512 for SINCERE and other contrastive losses. Various settings of temperature ($\tau$) and learning rate were evaluated, with the highest 1NN accuracy determining the final model parameters. The additional hyperparameter $\epsilon$ was searched over the values $[0.1, 0.25, 0.5]$ as in Barbano et al. (2023). The final models were trained on the entire training set, with evaluations on the test set reported in the main paper.

Transfer learning results used the SINCERE and SupCon ImageNet-100 models as frozen feature extractors for linear classifiers. This differs from the full model finetuning method used by Khosla et al. (2020) in order to more clearly determine the effects of the frozen embedding features. Our reported transfer learning accuracy results are more comparable to the transfer learning results by Chen et al. (2020a), although they opt for L-BFGS optimization without data augmentation instead of our choice of SGD optimization with random crops and horizontal flips. Linear classifier training was done with 100 epochs of stochastic gradient descent with 0.9 momentum, 0.0001 weight decay, and 128 batch size. Various learning rates were evaluated, with the highest classification accuracy on the 10% of the training set used as validation determining the final model parameters. The final models were trained on the entire training set, with evaluations on the test set reported in the main paper.

## K   Training Loss Comparison

The analysis of SupCon loss in Sec. 3.4 suggests that intra-class repulsion will increase the minimum loss value relative to SINCERE. Assuming uniform class frequencies and a fixed batch size, there are more loss terms responsible for intra-class repulsion as the number of classes decreases. Therefore there should be a larger difference between the minimums of SupCon and SINCERE loss as the number of classes decreases.

Table 4 clearly shows this in practice. SupCon and SINCERE losses have very similar values during the first training epoch, but the final training value for SINCERE loss is significantly lower than SupCon loss for all data sets due to the elimination of intra-class repulsion. Intra-class repulsion is most severe for the two class dataset CIFAR-2, with the final training loss value near the initial training loss value for SupCon. Moving to 10 classes in CIFAR-10 and then to 100 classes in the largest datasets, SupCon loss' final training loss value does decrease, but remains higher than SINCERE loss in each case.

| Pretraining Loss | CIFAR-10 | CIFAR-100 |
|---|---|---|
| SupCon | 95.78 | 75.96 |
| SINCERE | 95.93 | 75.86 |

Table 5: Accuracy of linear probing on test set. Differences are not statistically significant according to bootstrap interval analysis. **Takeaway:** It is unsurprising that SINCERE and SupCon perform similarly, given the fact that both methods induce clear separation between target and noise in Fig. 2, even though the margin of separation varies. This paper's core claim is that SINCERE leads to notably wider target-noise separation in Fig. 2 due to eliminating intra-class repulsion, not that this necessarily improves accuracy.

| | CIFAR-2 | | CIFAR-10 | | CIFAR-100 | | ImageNet-100 | |
|---|---|---|---|---|---|---|---|---|
| Training Loss | 1NN | 5NN | 1NN | 5NN | 1NN | 5NN | 1NN | 5NN |
| SupCon | 92.15 | 92.15 | 95.53 | 95.57 | **76.54** | **76.31** | 71.32 | 72.16 |
| $\epsilon$-SupInfoNCE | 92.08 | 92.03 | 95.97 | 96.01 | 75.52 | 75.44 | 70.52 | 70.95 |
| SINCERE | 92.75 | 92.55 | 95.88 | 95.91 | **76.23** | **76.13** | 71.18 | 71.36 |

Table 6: Accuracy of k-nearest neighbor classifiers on test set, using 32×32 resolution images. SINCERE's performance is essentially indistinguishable from SupCon. Results are boldfaced only when differences are statistically significant according to bootstrap interval analysis, showing $\epsilon$-SupInfoNCE performs worse than SINCERE and SupCon on CIFAR-100 but is otherwise indistinguishable. **Takeaway:** It is unsurprising that SINCERE and SupCon perform similarly, given the fact that both methods induce clear separation between target and noise in Fig. 2, even though the margin of separation varies. This paper's core claim is that SINCERE leads to notably wider target-noise separation in Fig. 2 due to eliminating intra-class repulsion, not that this necessarily improves accuracy.

## L   Supervised Classification Accuracy with Linear Probing

We fit a linear classifier on frozen embeddings and report test accuracy in Table 5. We do not see any significant differences: our SINCERE is indistinguishable from SupCon using the bootstrap significance testing. We suggest this should not be too surprising, because once target and noise have any modest separation, classifiers should be able to distinguish them accurately.

Previously, Khosla et al. (2020) suggested that linear classifiers were not any better than neighbor-based methods when evaluating SupCon. Plus neighbor methods avoid the need for any additional training, as highlighted in the appendix of Khosla et al. (2020): "We also note that it is not necessary to train a linear classifier in the second stage, and previous works have used k-Nearest Neighbor classification or prototype classification to evaluate representations on classification tasks."

Our reproduction confirms that linear and neighbor-based classifiers have similar accuracy on CIFAR-10 and CIFAR-100. Comparing Table 5 to Table 6, performance on each data set changes by less than a 0.6 percentage points in accuracy. Therefore k-nearest neighbor was chosen to be the primary evaluation on larger datasets like ImageNet-100.

## M   Supervised Classification Accuracy with Weighted k-Nearest Neighbors

We measure classification accuracy for each trained embedding model via a weighted k-nearest neighbor evaluation on the test set. As in Wu et al. (2018), cosine similarity is used to chose the nearest neighbors and to weight votes. See App. L for comparison to linear probing.

Table 6 reports accuracy using 1 and 5-nearest neighbors. The difference between accuracies for SINCERE and SupCon is not statistically significant in all cases, based on the 95% confidence interval of the accuracy difference (Foody, 2009) from 1,000 iterations of test set bootstrapping. The one statistically significant result shows that $\epsilon$-SupInfoNCE performs worse than SINCERE and SupCon on CIFAR-100.

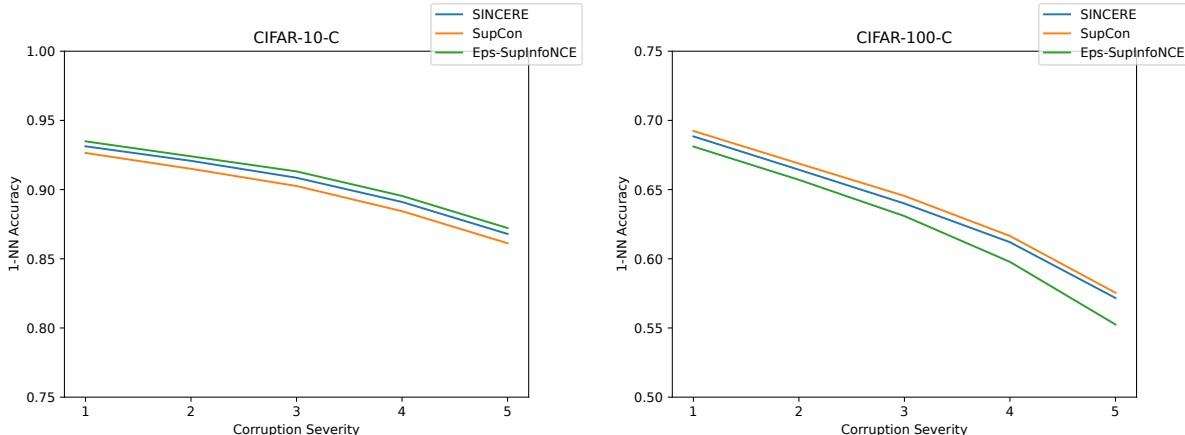

Figure 3: Change in 1-nearest neighbor accuracy across varying levels of corruption (Hendrycks & Dietterich, 2018) for SINCERE, SupCon, and $\epsilon$-SupInfoNCE. Differences in accuracy are **not statistically significant** based on the bootstrapped 95% confidence interval of the accuracy difference.

These results are surprising given how different the learned embedding spaces of the methods are. We hypothesize that this occurs because SupCon loss is still a valid nonparametric classifier. The model learns an effective classification function, but does not optimize for further separation of target and noise distributions like SINCERE does.

## N    Image Corruption Robustness

Figure 3 shows how each supervised contrastive method changes in 1-nearest neighbor accuracy across varying levels of corruption on CIFAR-10-C and CIFAR-100-C datasets (Hendrycks & Dietterich, 2018), all using ResNet-50 architectures. No statistically significant differences in accuracy are observed, aligning with previous results by Khosla et al. (2020) that observed little change in accuracy between cross entropy and SupCon losses.

## O    Learned Embedding Space by Class

Figure 4 shows the pairs from Figure 2 broken down by target and noise distributions for individual classes.

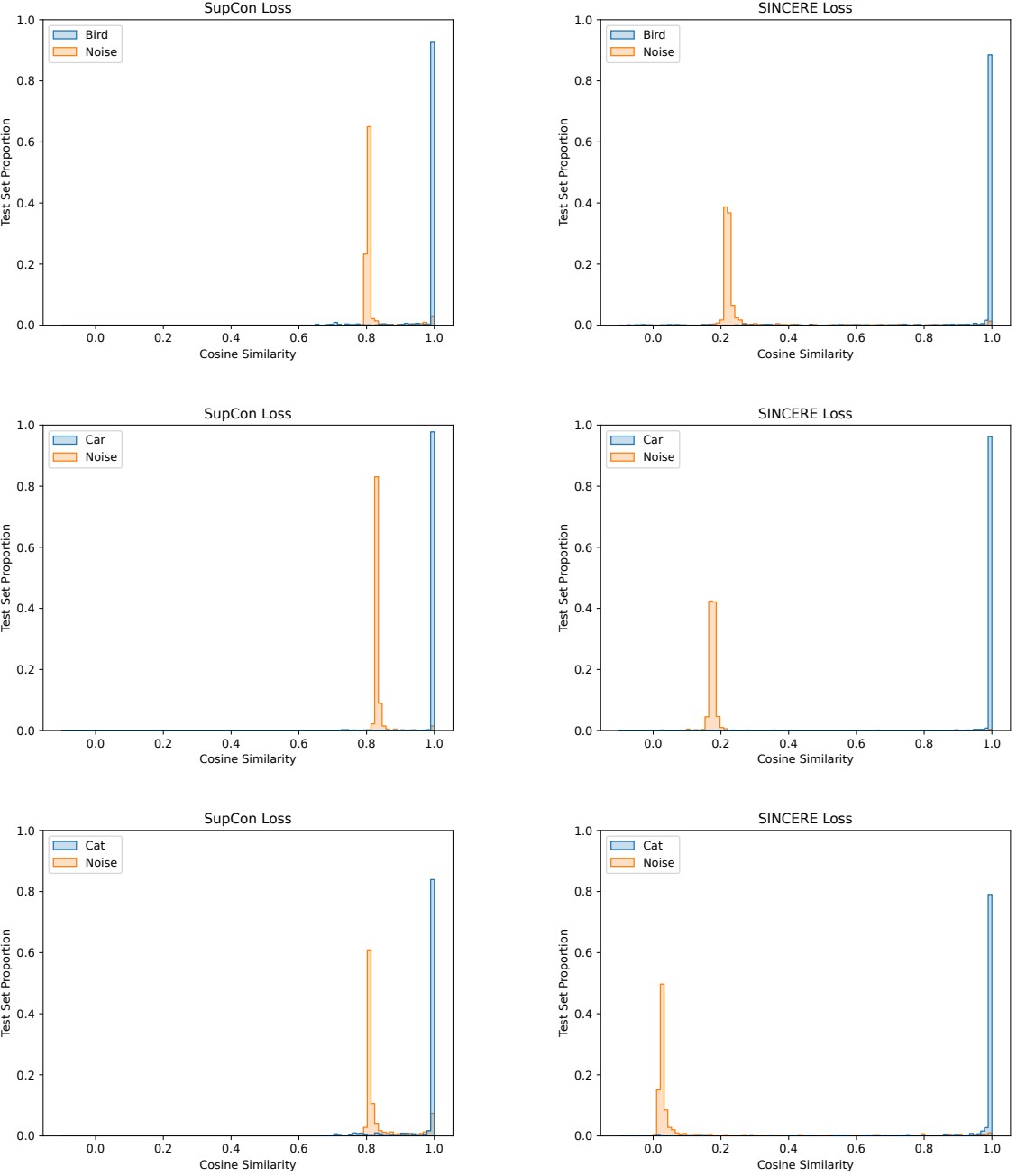

Figure 4: Histograms of cosine similarity values for CIFAR-10 test set nearest neighbors, comparing SupCon (left) and SINCERE (right). For each class, we plot the similarity of each test image with that class to the nearest target image in the training set as well as the nearest noise image in the training set. SINCERE loss maintains high similarity for the target distribution while lowering the cosine similarity of the noise distribution more than SupCon loss.

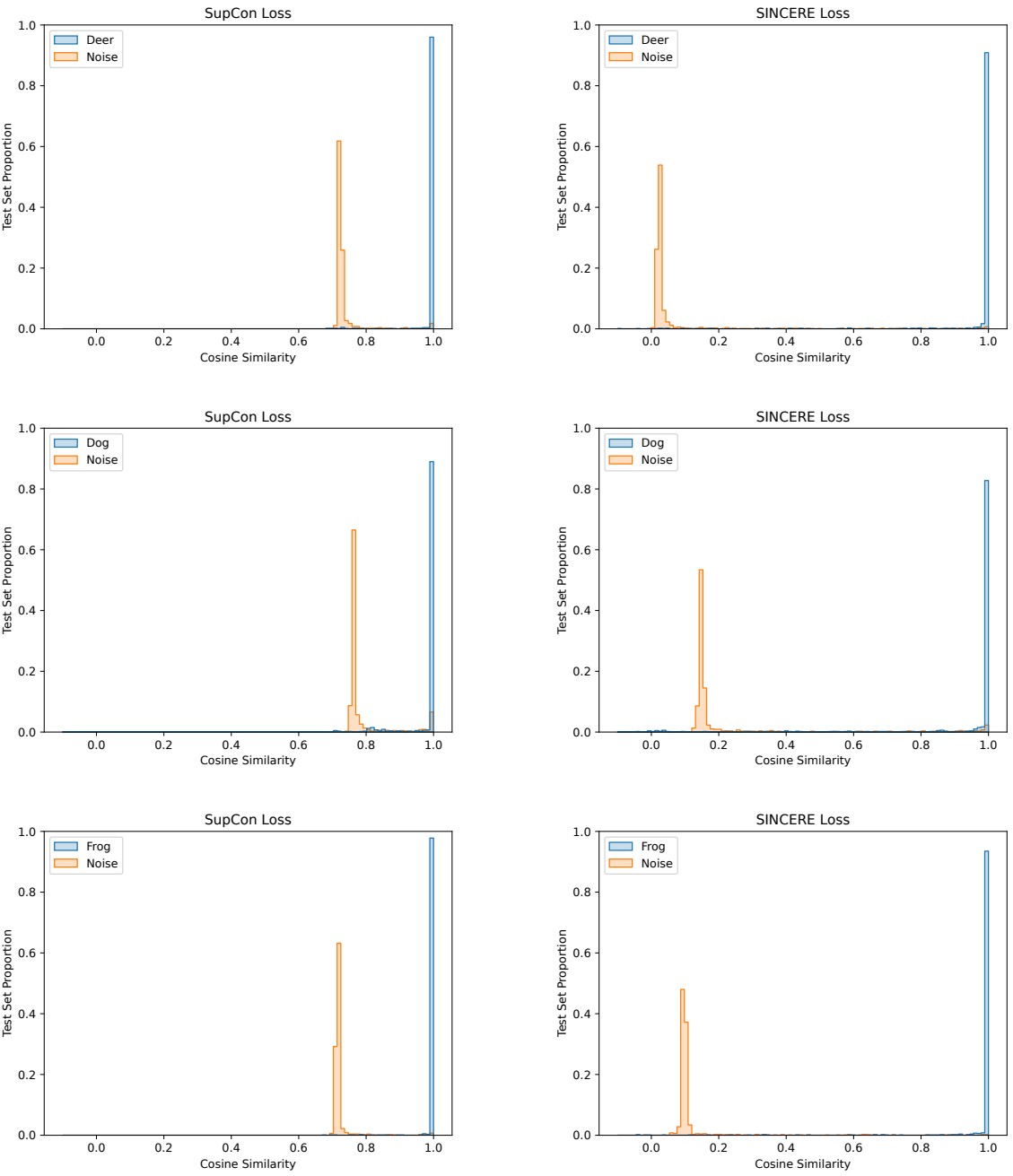

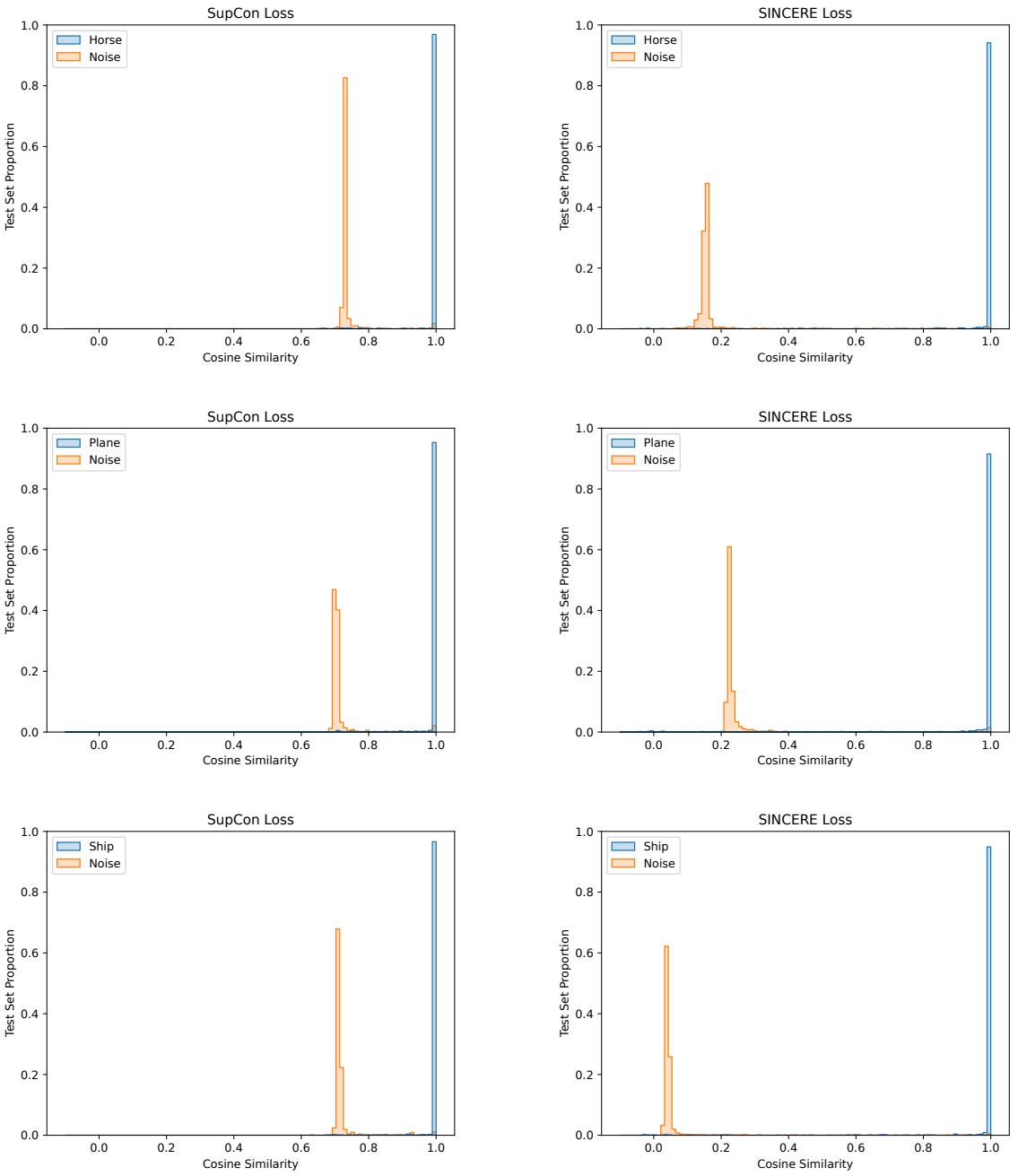

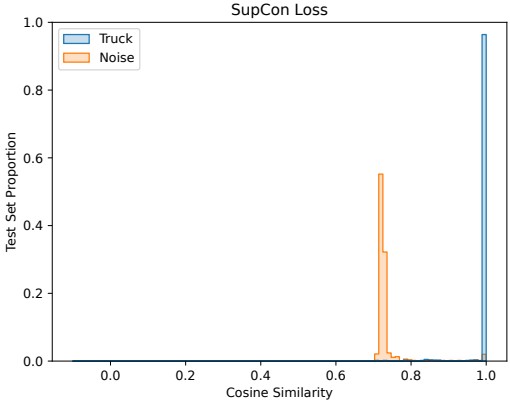 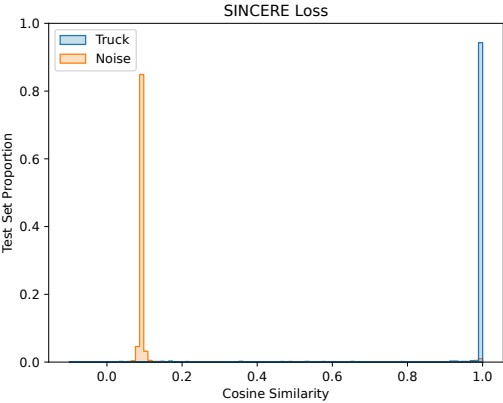

