# OpenReview forum: "SINCERE: Supervised Information Noise-Contrastive Estimation REvisited"
_TMLR — Rejected by TMLR_

### Review · Reviewer_YcmS · 2024-12-10

**Summary Of Contributions:**

The paper proposes SINCERE to address the intra-class repulsion in the SupCon loss. SINCERE eliminates this problem by ensuring that target samples are not treated as noise, leading to better target-noise separation and improved downstream performance in transfer learning tasks.

**Audience:**

Yes

**Claims And Evidence:**

Yes

**Requested Changes:**

see weakness

**Strengths And Weaknesses:**

**Strengths**：
1. The paper provides a solid theoretical framework for SINCERE loss.
2. The proposed method demonstrates clear empirical advantages.
3. The paper includes rigorous gradient analysis.

**Weaknesses**：
1. While the intra-class repulsion issue is central to the paper and serves as the foundation for the proposed algorithm, it is debatable whether this problem significantly impacts the performance of self-supervised methods. Many traditional self-supervised approaches treat samples from the same class as negatives in the embedding space and have proven to be effective in practice. From my experience, even when augmentations of a sample are treated as positives and all other samples (including those from the same class) are treated as negatives, these methods still perform well. This observation raises doubts about whether intra-class repulsion is truly a critical bottleneck in improving self-supervised algorithms.
2. If the paper aims to address the intra-class repulsion issue, the solution needs to be more thoroughly explained. From my understanding, SINCERE appears to simply omit a term from the SupCon loss that accounts for intra-class samples. This change, while effective, is not well justified in terms of theoretical or practical innovation. If there is additional nuance to the proposed method, it is not clearly articulated.
3. Although the method builds on and improves SupCon, focusing primarily on comparisons with SupCon and a few of its variations is insufficient to validate the effectiveness of the proposed algorithm. The paper lacks comparisons with state-of-the-art algorithms beyond SupCon. Additionally, the experiments are overly simplistic, relying solely on datasets like CIFAR and ImageNet-100 rather than more challenging benchmarks like ImageNet-1k. The choice of a single architecture, ResNet-50, limits the generalizability of the results. Furthermore, the evaluations are restricted to linear probing, omitting fine-tuning experiments, which are standard in the field.
4. The paper is challenging to read due to unclear structure and definitions. For example, the second paragraph of Section 2.3 is difficult to understand due to its complex and ambiguous definitions. Similarly, the derivation of the SINCERE loss in Section 3.1 is poorly connected across subsections. Specifically, it is unclear how the Bayesian formulation in Sections 3.1.1 and 3.1.2 leads to the SINCERE loss in Section 3.1.3. The lack of clear transitions and logical flow makes it hard to follow the reasoning.
5. Sections 3.2 and 3.3 include theoretical analyses, but their relevance to the design of the loss function is unclear. If these derivations neither influence the loss formulation nor contribute to the understanding of the core ideas, they might be better suited for the appendix.

---

> ### Author Response · Authors · 2025-01-09
>
> Thank you for your time and constructive feedback. We try to address each point raised.
>
> > W1: Why doesn’t intra-class repulsion hurt the performance of self-supervised contrastive learning?
>
> We are not trying to say that *self-supervised* contrastive learning (SSCL), like InfoNCE, suffers from intra-class repulsion. Our focus in this paper is about how *supervised* contrastive learning using SupCon suffers from intra-class repulsion.
>
> It is true that SSCL includes samples from the same semantic class (e.g. other “dog” images) as negatives, but this is because SSCL is solving a different problem than SINCERE or SupCon. Under SSCL’s instance discrimination problem, semantic classes are not the focus because such labels do not exist. Instead, the “classes” of interest are the instances themselves (z_S and z_p represent one particular dog, other embeddings represent other individual instances). InfoNCE loss under this definition does not have conflicting definitions of target and noise distributions as found in SupCon (Figure 1). SupCon’s inconsistent definition of target and noise is what fuels intra-class repulsion, so similar effects are not observed with the consistent definitions of SSCL. SINCERE is bringing the consistency of target and noise distributions found in SSCL to supervised learning, eliminating intra-class repulsion.
>
> To see that intra-class repulsion is a real problem, we encourage inspection of the histograms in Fig 2, which show that SupCon really doesn’t separate target and noise classes as well as possible, while our SINCERE makes big improvements.
>
> > W2: What is the justification for SINCERE?
>
> We show how the theoretical motivation for InfoNCE (Sec. 3.1.1) can be generalized for supervised learning (Sec. 3.1.2) and used to motivate SINCERE (Sec. 3.1.3). This theory allows us to show that the SINCERE loss is “ideal” for the assumed supervised setting, in the sense that a neural-net-based target-noise classifier that minimizes SINCERE loss will be as good as a classifier developed by someone with full knowledge of the true generative model (Sec. 3.1.4).
>
> In terms of practical effects, the change from SupCon to SINCERE corrects the intra-class repulsion found in SupCon’s gradients (Sec. 3.4) which improves the separation of classes in the embedding space (Sec. 4.1) and transfer learning accuracy (Sec. 4.2).
>
> > W3: Experiments are limited in scope.
>
> We focus Section 4.1 on comparisons to SupCon because we focus there on the effects of our change in loss function, necessitating comparison to a similar loss function. Similarly we focus Section 4.2 on other supervised contrastive methods. As most papers simply utilize SupCon, we found $\epsilon$-SupInfoNCE to be the only other comparable supervised contrastive method in the current literature. If you have other “state-of-the-art” losses in mind designed for the supervised contrastive setting, we’d appreciate an explicit pointer.
>
> In response to your and Reviewer nwh7’s request to study different architectures, we are currently planning on evaluating one additional architecture, ResNet-200, during the response period. Unfortunately, experiments are currently delayed due to scheduled maintenance of our institution’s GPU cluster, which is completely offline for the next 72 hours. We currently expect to deliver these results next week.
>
> To look at larger datasets like ImageNet-1k, we would require time beyond the allotted two week response period to complete such experiments. We would be willing to conduct those experiments if given additional time, but expect that it would take a month of compute time even when reducing the training time from 800 to 200 epochs.
>
> While finetuning is often used and would likely improve downstream classifier performance, we chose to use linear probing in our experiments because our goal is to evaluate the efficacy of the embedding functions learned by contrastive methods. Finetuning would change these embedding functions, making it less clear whether accuracy differences are attributable to the effects of the loss used for contrastive pretraining or variations that occurred during finetuning (which may not depend on SINCERE at all). We thus disagree that fine-tuning should be a “standard” evaluation for papers whose primary contribution is a new supervised contrastive loss.
>
> > W4: Clarity of Section 2.3
>
> We would appreciate some more specific feedback on how the second paragraph of Section 2.3 could be improved. Would a figure showing how a dataset is decomposed into the described sets help here?
>
> > W4: Clarity of Sec 3.1.1-3.
>
> There is a transition from a purely Bayesian model in Sections 3.1.1 and 3.1.2 to a neural network model approximation of those models in Section 3.1.3. We attempt to ease this transition through the parallels in Equations 8 and 9 before introducing the practical implementation in Equation 12. Would more explicitly highlighting this relationship in Section 3.1.3 improve the logical flow?

---

> ### Author Response · Authors · 2025-01-09
>
> > W5: Moving Sections 3.2 and 3.3 to Appendix.
>
> Sec. 3.2 contains a key theoretical contribution: a bound that relates our proposed SINCERE loss to a fundamental property of the target-noise contrastive task: the symmeterized KL divergence between the target class PDF and noise class PDF. The InfoNCE paper (van den Oord et al) offers a similar bound in their main paper, and it has spawned tons of valuable follow-up work cited at end of Sec 3.2. We wish to keep some form of this bound in the main paper, but we will consider moving Sec 3.2 after the current 3.3 and 3.4 and moving some details to the Appendix.
>
> Section 3.3 introduces the practical implementation of SINCERE loss used in experiments, so we believe that it is necessary in the main paper in some form. We’ll consider  whether some details could be moved to the Appendix.

---

### Review · Reviewer_nwh7 · 2024-12-28

**Summary Of Contributions:**

The paper enhances the separation between classes in the embedding space by modifying the supervised contrastive loss function, Supervised InfoNCE REvisited (SINCERE).  The idea of SINCERE is to eliminate the target class instances from the denominator of the InfoNCE loss function, which in turn eliminates the intra-class repulsive.

**Audience:**

No

**Claims And Evidence:**

No

**Requested Changes:**

1-The author states that eliminating intra-class repulsive allows for better class separation, which in turn enhances representation learning. Can the author explain why making a firm separation between classes in the embedding space is useful for representation learning? Note that the performance has not significantly improved, and in some cases, SupCon performs better than SINCERE, as shown in Tables 5 and 6.

2-We encourage the author to conduct more experiments on different downstream tasks, such as fine-tuning and object detection, and compare them with SupCon. We also encourage the author to use different measures to evaluate the approach, such as robustness (see [1] Figure 3).  This will illustrate the impact of the introduced approach on representation learning.

3-We recommend the author do ablation studies for the methodology to show the consistency of the approach, especially for the batch size, which plays an important role in contrastive learning (see [1] Figure 4) for more ideas about the ablation studies.

4-We are concerned about whether finding the target class instances and eliminating them from the denominator adds more time complexity or not. Thus, we encourage the author to use FLOPs to compare the two approaches (i.e., SupCon and SINCERE).

5-Regarding scalability, SupCon performance improves when using a large Dataset (ImageNet) and large architectures (e.g., ResNet 101 and ResNet 200). We are concerned whether the proposed idea can perform the same on a larger scale.


[1] Khosla, Prannay, et al. "Supervised contrastive learning." Advances in neural information processing systems 33 (2020): 18661-18673.

**Strengths And Weaknesses:**

**Paper strength:**

1-	The author describes the motivation of the paper clearly.

2-	The comprehensive literature review provides a strong foundation for the presented work and demonstrates a deep understanding of the existing research landscape.

3-	The author presents the core idea, making it easily understandable.

**Paper weakness:**

1-	According to the results in Tables 5 and 6 on pages 26 and 27, while the SINCERE approach successfully increased the distance between different classes representation in the embedding space, this improved separation did not translate into a significant performance improvement for the trained model. This concerns me if we need to resort to this approach.

2-	Limit evaluation for the approach.

3-	Limited ablation study.

4-	The paper does not discuss the computational complexity or scalability of the proposed approach in detail, which could be a concern for large-scale applications.

---

> ### Author Response · Authors · 2025-01-09
>
> Thank you for your careful reading and constructive feedback. We try to address each point raised.
>
> > W1/RC1: Why is making a firm separation between classes in the embedding space useful for representation learning?
>
> Separating classes in embedding space is the stated goal of supervised contrastive learning. From the abstract of the SupCon paper by Khosla et al, the goal is to create embeddings where “Clusters of points belonging to the same class are pulled together in embedding space, while simultaneously pushing apart clusters of samples from different classes.” We argue that while our work and SupCon agree on this goal, SupCon loss does not achieve it as well as our SINCERE loss, as evidenced by the histogram plots in our Fig 2.
>
> Favoring the widest possible separation of classes in the embedding space is preferred based on the maximum margin principle, a principle that underpins the representation learning of many popular methods in ML over decades. There are intuitive arguments for favoring large margins (see Sec 17.3.1 of Murphy’s Probabilistic Machine Learning textbook). There are also more formal arguments related to guarantees about how well a classifier might do on test data. Many works [a] have proven generalization bounds that scale “inversely polynomial with the margin size” [b], in other words: these bounds suggest that wider margins between classes imply lower classification error on never-before-seen data drawn from the same distribution as the train set.
>
> [a] Colin Wei and Tengyu Ma. Improved sample complexities for deep networks and robust classification via an all-layer margin. In International Conference on Learning Representations (ICLR), 2020.
> [b] Glasgow et al. 2023. https://arxiv.org/pdf/2206.07892
>
> > RC1: “performance has not significantly improved” as measured by standard classification accuracy.
>
> It is true that using the measure of test accuracy, our method is not statistically different (not better but also not worse) than SupCon. This shouldn’t be too surprising: looking at the separation histograms in Fig 2, using a nearest neighbor classifier on SupCon embeddings would clearly have high accuracy at distinguishing target from noise classes.
>
> We argue that SINCERE improves on SupCon by maintaining that accuracy while being better justified theoretically and more in line with the maximum margin principle. This improvement is quite clear in Fig 2 and Table 1: SupCon’s median separation of target and noise is significantly lower than our method’s across all four tested datasets. We believe that this drives the accuracy gains in transfer learning that we report in Table 2.
>
> > RC1: “in some cases, SupCon performs better than SINCERE, as shown in Tables 5 and 6.”
>
> Please use caution before thinking either table suggests SupCon is somehow truly better. In these tables results are boldfaced “only when differences are statistically significant according to bootstrap interval analysis”. There is no case in either table 5 or table 6 where SupCon is statistically significantly better than SINCERE. Any apparent numerical difference is not significant; the two losses on these tasks appear indistinguishable.
>
> > W2/RC2: Request to “conduct more experiments on different downstream tasks, such as fine-tuning and object detection, and compare them with SupCon”.
>
> We argue that our current experiments better focus on our stated goals: evaluating the quality of representations directly learned by our loss versus competitor loss functions.
>
> While full fine-tuning of all layers is often used in the broader ML literature and would likely improve downstream classifier performance, we chose to use kNN and linear probing with fixed embeddings in our experiments because it better aligns with our specific  goal: evaluate the efficacy of the embedding functions learned by contrastive methods. Finetuning would change these embedding functions, making it less clear whether accuracy differences are attributable to the effects of contrastive training or variations that occurred during finetuning (which may not depend on SINCERE at all).
>
> We view object detection as quite out-of-scope, as it would require specialized losses that differ from our focus on classification. The close competitor methods we compare to, like SupCon, also do not pursue object detection in their original papers.
>
> > RC2 and RC3: Robustness and ablation experiments.
>
> We are currently pursuing experiments for robustness with CIFAR-10/100-C and an ablation study on batch size. We currently expect to deliver these results next week.

---

> > ### Comment · Reviewer_nwh7 · 2025-01-09
> >
> > Thank you for the author's response.
> >
> > a)- As you know, we are using different downstream tasks as a proxy for the useful representation learning for the model. In Table 5, SupCon outperforms SINCERE on the CIFAR-100 dataset. Also, SupCon performs better than SINCERE  on two datasets, CIFAR-100 and ImageNet-100, as shown in Table 6. It is true that Separating classes in embedding space is the stated goal of supervised contrastive learning, and in Figure 2, the SINCERE added more separability but my question: **Could you please clarify why I need the extra separability by SINCERE, since this separation is not adding to the model performance (i.e., linear probing and KNN which we used to evaluate the two approaches)?**
> >
> > b)- we also recommend comparing the two approaches in terms of time complexity. If SINCERE is doing better than SupCon, this point will be added to the SINCERE and vice versa.

---

> > > ### Author Response · Authors · 2025-01-10
> > >
> > > > a) why need the extra separability by SINCERE, since this separation is not adding to the model performance?
> > >
> > > We do seem to have a choice between two losses with indistinguishable accuracy, as judged by significance testing in Tables 5 and 6.
> > >
> > > The advantages of SINCERE in our view are:
> > >
> > > * SINCERE yields significantly better *transfer learning* (see Table 2), with better accuracy on 4 out of 6 datasets.
> > > * SINCERE yields better separability, which aligns with the max margin principle.
> > > * SINCERE has a cleaner conceptual foundation. We can explain why each term in the loss exists, based on the derivation in Sec. 3, and how it contributes to the stated goals of supervised contrastive learning.
> > >
> > > There's not really a downside to SINCERE that we can see. Better transfer learning on average, a strong conceptual foundation, and equally strong empirical performance.
> > >
> > >
> > > > RE reviewer's references to Table 5 and 6
> > >
> > > Again, please be careful interpreting either Table 5 or Table 6 as indicating that "SupCon performs better than SINCERE" in some cases. The captions of these tables clarify that statistical testing did not indicate any significant differences between SINCERE and SupCon on any dataset. From this significance testing, we conclude the two methods are **indistinguishable**. Any apparent minor numerical differences are not meaningful.
> > >
> > > > b)  we also recommend comparing the two approaches in terms of time complexity
> > >
> > > App E clarifies that the two approaches have the same runtime complexity, as was mentioned in our earlier response labeled "W4/RC4". Please let us know if we can clarify further.

---

> ### Author Response · Authors · 2025-01-09
>
> > W4/RC4: Paper does not discuss computational complexity and unclear if the denominator calculation is more computationally expensive.
>
>
> See our submission’s Appendix E, which shows that both SINCERE and SupCon have indistinguishable big-oh complexity for both runtime and memory.
>
> We have revised Appendix E to directly address your question about the cost of the denominator calculation after the calculation of pairwise similarities in $O(N^2 D)$ time. In short, the calculation of the up to $N^2$ unique denominators is done in two $O(N^2)$ operations: calculating one "base" denominator for each index $S$ that excludes the similarities between $S$ and the potential partners $p$ ($N$ entries with $N$ operations, $O(N^2)$ time), then combining that with the $S$ and $p$ similarities ($N \times N$ entries with constant operation, $O(N^2)$ time). The calculation of the denominator terms is therefore dwarfed by the $O(N^2 D)$ time required by both methods to compute the pairwise dot products. Please see the revised text for a more thorough explanation.
>
> Overall, the most practical thing we can say about runtime is that the time to compute the embeddings of a typical batch (100s of images) far outweighs the cost of computing either the SINCERE or the SupCon loss given the embeddings.
>
>
> > RC4: Request for FLOPs comparison between SupCon and SINCERE
>
> We modified the code included in our supplement to utilize two possible FLOP counters, one from PyTorch [c] and another from fvcore [d]. Both counters are limited in the kinds of operations they are able to count. For example, PyTorch’s counter focuses on matrix multiplies but cannot track FLOPs through exp or log operations, among others. Thus, automatic FLOP counting should be viewed as approximate. Both counters report that Khosla’s implementation of SupCon and our implementation of SINCERE are exactly equivalent in terms of FLOPs for both the forward and backward passes. For example, 512 64-dimensional embeddings evenly split across 8 classes leads to both methods requiring 67 million FLOPs for the forward pass and 134 million FLOPs for the backward pass. This aligns with our finding that there is no significant difference in wall clock speed for such batches.
>
> fvcore’s FLOP counter does report operations that require FLOPs but are not supported. This shows that SINCERE does require an extra logsumexp operation to handle the variable denominators, but the remaining differences are minor (e.g. slightly different functions are used to construct boolean masks).
>
> [c] https://pytorch.org/tnt/stable/utils/generated/torchtnt.utils.flops.FlopTensorDispatchMode.html
>
> [d] https://github.com/facebookresearch/fvcore/blob/main/docs/flop_count.md
>
> > RC5: Can SINCERE scale up to a larger Dataset (ImageNet) and large architectures (e.g., ResNet 101 and ResNet 200)?
>
> In response to your and Reviewer YcmS’s request to study different architectures, we are currently planning on evaluating one additional architecture, ResNet-200, during the response period. Unfortunately, experiments are currently delayed due to scheduled maintenance of our institution’s GPU cluster, which is completely offline for the next 72 hours. We currently expect to deliver these results next week.
>
> Based on our big-oh runtine analysis in App E, we don’t see any reason to expect SINCERE would not scale up as well as SupCon. However, to complete experiments on larger datasets like ImageNet-1k, we would require time beyond the allotted two week response period. We would be willing to conduct those experiments if given additional time, but expect that it would take a month of compute time even when reducing the training time from 800 to 200 epochs.

---

### Review · Reviewer_z1fx · 2025-01-04

**Summary Of Contributions:**

The authors propose modifying the SupCon framework by removing the intra-class repulsion in the SupCon InfoNCE loss.

**Audience:**

Yes

**Claims And Evidence:**

Yes

**Requested Changes:**

Check "Weakness" section

**Strengths And Weaknesses:**

**Strengths**:

- The author proposes an improved modification to SupCOn by eliminating the intra-class repulsion from the Supervised InfoNCE loss function.
- The motivation is subtle but pretty valid.

**Weakness**:
- The authors provide the linear probing results. However, it is not clear, how it differs from transfer learning in this case, as it is not mentioned if the frozen embeddings were obtained after training on ImageNet or CIFAR dataset.
- The authors could also provide semi-supervised fine-tuning performance.
- The results of linear probing and semi-supervised fine-tuning could be provided before the transfer learning results.
- The term in the numerator of the SINCERE loss is present in the denominator. Does that not also exert a repulsion on the samples in the same class?
- From my personal experience, an A100 GPU with 40 GiB of memory is sufficient to run experiments with 224x224 dimensions with a batch size of 256. It would enrich the paper, if authors can provide results on large-scale datasets instead of resizing, as is often the convention.

---

> ### Author Response · Authors · 2025-01-09
>
> Thank you for your time and constructive feedback. We try to address each point raised.
>
> > “The authors provide the linear probing results. However, it is not clear, how it differs from transfer learning in this case…” “The results of linear probing and semi-supervised fine-tuning could be provided before the transfer learning results.”
>
> To clarify, the linear probing results in Table 2 *are* our transfer learning results. They are not separate from transfer learning as linear probing is the transfer learning method we used.
>
> To clarify what we mean by “linear probing”, we first trained an embedding network by optimizing SINCERE loss (or competitor loss methods) on the ImageNet-100 dataset. This embedder’s weights are then frozen and not updated further. We create a classifier for a target dataset (say the 37-class Pets dataset) with two parts: first, the frozen embedding network, and second, a freshly-initialized linear-softmax output layer that maps from embeddings to class probabilities. This is a fairly standard way to do transfer learning that is more affordable that “full fine-tuning” of all the embedding network’s layers for the target task.
>
> > Were “frozen embeddings … obtained after training on ImageNet or CIFAR dataset”?
>
> In Sec 4.2, we state “We choose classification on ImageNet-100 (Tian et al., 2020a; Chun-Hsiao Yeh, 2022) as our source task to evaluate transfer learning”, meaning that we used ImageNet-100 as the source dataset for training representations. This is also clarified in the caption of Table 2, which states we do “transfer learning from ImageNet-100”.
>
> > Request for “semi-supervised fine-tuning”
>
> We agree that semi-supervised fine-tuning could provide additional empirical results. However, as semi-supervised fine-tuning has not been explored in previous contrastive learning works (especially in the most similar works by Khosla et al. 2020 and Barbano et al. 2023), to our knowledge, we believe that such experiments are beyond the scope of our paper. We would appreciate any pointers to papers on semi-supervised fine-tuning of supervised contrastive learning, if available, as we are not familiar with that literature.
>
> > “The term in the numerator of the SINCERE loss is present in the denominator. Does that not also exert a repulsion on the samples in the same class?”
>
> No, see the analysis of gradients in Sec 3.4. Instances indexed by p and S in our notation always belong to the same class. Inspecting Eq 13, we can see that p’s embedding will be pulled toward the location of $z_S$, because the first term of the gradient is always a negative value multiplied by $z_S$ (and the second term is only using $z_S \cdot z_p$ to inform the weight of the repulsion from noise embeddings). Recall the gradient points in the direction of steepest ascent of the loss. To lower the loss, iterative gradient *descent* will take a step opposite this gradient, meaning toward $z_S$. Thus, S and p will be attracted, not repelled.
>
> > “It would enrich the paper, if authors can provide results on large-scale datasets instead of resizing, as is often the convention.”
>
> Have you been able to use that resolution and batch size for contrastive learning methods? Our experience is that 40 GiB is sufficient for the forward pass, but will run out of memory during the backward pass. This is due to the fact that these contrastive losses do not use stop gradients, so each embedding’s gradient will be affected by most of the loss calculations for the batch through its presence in the denominator. This differs from cross entropy loss where a single loss calculation is all that affects the gradient of a given embedding.
>
> This issue has not been explicitly discussed in previous work, to our knowledge, but work such as MoCo v3 [a] utilizes at most 32 images per GPU with 224x224 resolution images for their SimCLR loss variant on ResNet-50.
>
> [a] Chen, Xinlei, Saining Xie, and Kaiming He. "An empirical study of training self-supervised vision transformers." Proceedings of the IEEE/CVF international conference on computer vision. 2021. https://arxiv.org/pdf/2104.02057

---

> ### Comment · Reviewer_z1fx · 2025-01-09
> **Reply to Author's Comment**
>
> > To clarify what we mean by “linear probing”, we first trained an embedding network . . . that “full fine-tuning” of all the embedding network’s layers for the target task.
>
> Yes, linear probing in SSL is pretty common. I only asked if the authors could add information on which dataset the embedding network was trained, as it was not easily understandable.
>
> > However, as semi-supervised fine-tuning has not been explored in previous contrastive learning works (especially in the most similar works by Khosla et al. 2020 and Barbano et al. 2023), to our knowledge, we believe that such experiments are beyond the scope of our paper.
>
> It may be that previous works have not explored semi-supervised fine-tuning. But that should not be a strong reason not to explore something. That will be detrimental to the research progress if this logic is applied. Semi-supervised fine-tuning can give more insights into the performance of the proposed framework, and may also help the authors establish the superiority of their framework when it is observed that the proposed framework _does not outperform_ the existing works (Khosla et al. 2020 and Barbano et al. 2023) significantly in kNN and Linear Probing accuracy.
>
> > No, see the analysis of gradients in Sec 3.4. Instances indexed by p and S in our notation always belong to the same class. Thus, S and p will be attracted, not repelled.
>
> If you carefully look at the expression in Eqn. 38, and expand it further by separating the numerator and denominator, you can see what I was talking about. If the expression is expanded, then a term involving a positive pair will be involved in such a way that it can be seen as exerting a repulsive force on one another. However, this pair will come inside the $\log \sum \exp$ term. If this term is removed, then the loss will become similar to the DCL loss (Yeh et al., 2022) for the supervised case.
>
> > Have you been able to use that resolution and batch size for contrastive learning methods?
>
> I can suggest the authors use *fp16* precision during pre-training. I have experience training contrastive SSL methods on a 24GB GPU with batch size 128. Also, the lightly-ai benchmark states that they can train on ImageNet using batch size 256 using 2 GPUs (4090), which equates to 48GB using '16-mixed' precision (default). Based on these, I think it will be possible to run with batch size 256 on the available GPU. However, the authors can at least try with batch size 128 and fp16 precision for 100 epochs.
>
> References:
>
> [1] Yeh, CH., Hong, CY., Hsu, YC., Liu, TL., Chen, Y., LeCun, Y. (2022). Decoupled Contrastive Learning. In: Avidan, S., Brostow, G., Cissé, M., Farinella, G.M., Hassner, T. (eds) Computer Vision – ECCV 2022. ECCV 2022. Lecture Notes in Computer Science, vol 13686

---

> > ### Author Response · Authors · 2025-01-10
> >
> > > “add[ing] information on which dataset the embedding network was trained [on]”
> >
> > We hope that highlighting where we list the source dataset used for pretraining for transfer learning has clarified this sufficiently. We are happy to revise the text if there is something unclear about those statements.
> >
> > > “Semi-supervised fine-tuning can give more insights into the performance of the proposed framework”
> >
> > We do agree with this point, but do not think we have the time to expand our results to include this new task during the revision period in addition to the other additional experiments jointly requested by the other reviewers. We hope you understand that there’s only so much we can do in the brief response period.
> >
> > > “the proposed framework does not outperform the existing works significantly in kNN and Linear Probing accuracy”
> >
> > SINCERE does statistically significantly outperform existing works in linear probing accuracy for transfer learning, as we highlight in Table 2.
> >
> > > “If [Eq. 38] is expanded, then a term involving a positive pair will be involved in such a way that it can be seen as exerting a repulsive force on one another. However, this pair will come inside the logsumexp term. If this term is removed, then the loss will become similar to the DCL loss (Yeh et al., 2022) for the supervised case.”
> >
> > First we would like to thank the reviewer for drawing our attention to App. D as it contained some notation that was not in line with the rest of the paper. This has been fixed in our latest revision, along with slight formatting changes to the equations for increased clarity.
> >
> > Consider the gradient of the DCL loss gradient with respect to $z_p$ (written in our notation)
> >
> > $$\frac{1}{\tau} \left( -z_{S} + \sum_{i \in \mathcal{N}} z_i \Big( \frac{ e^{z_i \cdot z_p / \tau}}{\sum_{i' \in \mathcal{N}} e^{z_{i'} \cdot z_p / \tau}} \Big) \right)$$
> >
> > in contrast to the gradient of SINCERE loss with respect to $z_p$
> >
> > $$\frac{1}{\tau} \left( z_S \Big(\frac{e^{z_S \cdot z_p / \tau}}{\sum_{j \in \mathcal{N} \cup \{S\}} e^{z_j \cdot z_p / \tau}} - 1 \Big) + \sum_{i \in \mathcal{N}} z_i \Big( \frac{ e^{z_i \cdot z_p / \tau}}{\sum_{j \in \mathcal{N} \cup \{S\}} e^{z_j \cdot z_p / \tau}} \Big) \right).$$
> >
> > Our previous response focused on the first term and showed why $p$ and $S$ will be attracted to one another due to this term.
> >
> > We now focus on the second logsumexp term that you highlight in your comment. With our revised equation formatting, we highlight two properties of this term. First, inside the sum, at each noise index $i$ there is a noise embedding $z_i$ that $z_p$ will be repulsed from and a scalar fraction creating the weight of that repulsion. Second, the distinct indexing variable in the denominator highlights that $z_S$ is only part of the weighting term. Therefore the second term of the SINCERE loss cannot create a repulsion from $z_S$.
> >
> > But why is $z_S$ missing from the outer summation in the second term (indexed with $i$ instead of $j$ above) when DCL uses indices over the same set in both of their sums? In our Eq. 39 there is such a term utilizing $z_S$ in the numerator. Eq. 40 separates this term from the summation over noise instances indexed with $i$ above. Eq. 41 does two things: bring the $-1$ multiplier into the summation and combine the two terms utilizing $z_S$ to create the first term of our final SINCERE gradient expression. Thus the term of concern in your comment is used to determine the weight of the attraction towards $z_S$. This attraction cannot become a repulsion (a positive weight) since the weight on $z_S$ will always be negative or 0. SupCon lacks this property (Section 3.4) and DCL instead uses a constant weight of $\frac{-1}{\tau}$ on $z_S$, not adjusting this weight based on the separation of the noise embeddings.
> >
> > This explanation of Eq. 39-41 has been added to App. D in our latest revision. Please let us know if you have further questions.
> >
> > > “authors can at least try with batch size 128 and fp16 precision for 100 epochs”
> >
> > We appreciate your insight on this topic. We will investigate whether this approach works on our cluster after completing the experiments jointly requested by the other reviewers.

---

> > > ### Comment · Reviewer_z1fx · 2025-01-15
> > > **Reply from Reviewer**
> > >
> > > > since the weight on $z_S$ will always be negative or 0
> > >
> > > So, the weight on $z_S$ is attenuated in your proposed framework. Isn't that harming your performance? The gradient of your loss with respect to $z_p$ indicates how the samples in a positive pair will behave in terms of spatial distribution in the feature space (I hope I am able to make myself clear here, please ask if not.)
> > >
> > > For example,
> > >
> > > $z^{t+1} = z^t - \frac{dz}{dt} = z^t - \frac{dL}{dt}\frac{dz}{dL} \implies \lVert z^{t+1} - z^t \rVert = \lVert \frac{dz}{dt} \frac{dz}{dL}\rVert \leq \lVert \frac{dz}{dt} \rVert \lVert \frac{dz}{dL}\rVert = c \lVert \frac{dz}{dL}\rVert = c\lVert \frac{\tau}{z_s (p_{sp}-1) + \sum_{i\in{\mathcal{N}}} p_{ip} z_i} \rVert$
> > >
> > > Can you say anything about the rate of convergence for your framework? Since you already show things like the difference in the distribution of samples in each class vs. noise, you can ignore this comment if you want.

---

> > > > ### Author Response · Authors · 2025-01-15
> > > >
> > > > The way we interpreted your questions is roughly "How does the varying weight on $z_S$ in the gradient affect the performance and rate of convergence for SINCERE as compared to other methods such as DCL, which use a constant weight on $z_S$?" (Please let us know if our interpretation is incorrect in some way.)
> > > >
> > > > We consider how the weights on vectors differ between SINCERE and DCL losses to provide intuition for the differing behavior without currently considering the complexities of, for example, the parts of gradients in opposite directions cancelling out.
> > > >
> > > > With the SINCERE loss, the magnitude of the weight on the target distribution vector and the magnitude of the weights on the noise distribution vectors will sum to 1. This means that the loss will dynamically adjust the weight of these terms to more dramatically update embeddings that are not satisfying the objective. For example, if the target distribution similarity is already nearly maximized, as we see occurs in Figure 2, then the weight will be focused on the noise distribution vectors and the gradient will be dominated by positive terms.
> > > >
> > > > In contrast, DCL fixes the magnitude of the weight on the target distribution vector to 1 and lets the magnitude of the weights on the noise distribution vectors sum to 1. Therefore the target distribution update will always have the same weight and only the noise distribution vectors will have their weights dynamically adjusted. Thus the target distribution similarity does not effect the balance of the weight of negative and positive terms in the gradient.
> > > >
> > > > Relating this to rate of convergence, the overall weight on vectors in SINCERE is 1 while for DCL it is 2. If we consider SINCERE with double the learning rate of DCL and with batches constructed such that the magnitude of the weight on the target distribution vector and the sum of the magnitude of the weights on the noise distribution vectors are even (0.5 target distribution, 0.5 across the noise distribution) then the gradient descent updates would be equivalent. Therefore we believe that, given an appropriate choice of learning rate for each method, SINCERE and DCL do not have clear advantages over each other in terms of rate of convergence. The effects of each method on downstream performance is also unclear, especially with the additional context that DCL is designed as a self-supervised objective and a supervised extension has not be investigated to our knowledge.

---

> ### Comment · Reviewer_z1fx · 2025-01-16
> **Reply from Reviewer**
>
> >  the overall weight on vectors in SINCERE is 1 while for DCL it is 2
>
> I am not sure if that is correct. I think the overall weight of the vectors in both is 0.
>
> In SINCERE,
>
> weight on positive pair sample $z_S = p_{Sp} - 1$
>
> sum of weight on negative pair samples $z_i = \sum_{i \in \mathcal{N}}p_{ip}$
>
> Sum of weights = $\sum_{i \in \mathcal{N}}p_{ip} + p_{Sp} - 1 = \sum_{i \in \mathcal{N}\cup S}p_{ip} -1 = 1-1 = 0$
>
> In DCL,
>
> weight on positive pair sample $z_S = - 1$
>
> sum of weight on negative pair samples $z_i = \sum_{i \in \mathcal{N}}p_{ip} = 1$
>
> Sum of weights = $1-1 = 0$
>
> While the choice of learning rate is generally related to a hypothetical Lipschitz constant, can you say that you can simply double it in that case?
>
> >  Thus the target distribution similarity does not effect the balance of the weight of negative and positive terms in the gradient
>
> It is of my opinion that, the noise distribution will be implicitly affected by the positive pair samples. As you cannot directly dissociate the negative and positive samples and there is always an anchor sample, the effect of attraction on the positive pair samples will implicitly affect the repulsive forces on the negative pair samples in the process of optimization.

---

> > ### Author Response · Authors · 2025-01-22
> >
> > We agree that the overall weight sum is 0; we meant to refer to the total magnitude of the weights as discussed in the previous paragraphs.
> >
> > The case we highlighted with the doubled learning rate and $p_{Sp} = 0.5$ would produce identical gradient descent updates for both SINCERE and DCL, so the optimization process would be identical.
> >
> > We agree that the differing learned embedding distributions for target and noise distributions will result changes to the *vector* terms of the gradient. We were making a statement about the *scalar* weight terms, which for DCL has constant magnitude for the positive and negative weights (that is the $1$ and $-1$ you highlight as opposed to the dependence on similarities in SINCERE).
> >
> > We hope that this discussion has highlighted the differences between DCL and SINCERE. We could revise Section 3.5.1 to expand our discussion of DCL with the topics discussed, if desired by the reviewers.

---

> > > ### Comment · Reviewer_z1fx · 2025-01-25
> > > **Reply from Reviewer**
> > >
> > > > We were making a statement about the scalar weight terms, which for DCL has constant magnitude for the positive and negative weights (that is the and you highlight as opposed to the dependence on similarities in SINCERE).
> > >
> > > How do DCL has constant weight for both positive and negative? If the authors are only looking at the summed weights, then both DCL and SINCERE could be said to have constant weights.
> > >
> > > In both DCL and SINCERE, the weights are dependent on similarities. Except, in SINCERE the similariy of the positive pair is also included.
> > >
> > > I think this discussion has not highlighted the differences between DCL and SINCERE, in terms of how the optimization process is affected due to the differences in the gradient formulation.
> > >
> > > Furthermore, discussion based on the the case $p_s =0.5$ is not actually a good way to discuss how the entire optimization will happen, just because it fits the narrative of double learning rate.
> > >
> > > Lastly, I don't think the authors should add this discussion to the paper, as it won't be a good addition with so many incorrect statements made by the authors to show their method better than DCL. While I am not advocating for DCL to be the best method there is, a fair discussion would add more value to the paper.

---

> > > > ### Author Response · Authors · 2025-01-29
> > > >
> > > > We appreciate that we have different understandings of the subtleties DCL and thank you for your discussion. We want to highlight that DCL is self-supervised learning method and has not been adapted for supervised contrastive learning, which is the purpose of SINCERE.

---

### Author Response · Authors · 2025-01-25
**New revision with expanded experiments now available**

Dear reviewers and action editor,

We've just uploaded a revision that adds many new experiments suggested by reviewers. Please browse the latest PDF, looking for red text to see our latest changes (mostly in Sec 4 and Sec 5).

As you make final decisions, we hope reviewers can keep in mind the two central experimental claims of our paper (see top of Sec. 4):

1) SINCERE repairs the intra-class repulsion issue of SupCon, obtaining better target-noise separation

2) SINCERE offers representations that generalize well to new tasks via transfer learning.

We also hope reviewers appreciate the efforts we've put into assessing whether differences in numerical results are *statistically significant*. Previous evaluations in the original SupCon paper did not do such testing. We think such testing is a crucial best practice going forward to ensure the ML literature provides high-quality and reliable scientific claims.

Please let us know if we can clarify anything further.

Sincerely,
Authors of SINCERE

---

> ### Author Response · Authors · 2025-01-25
> **Change log in latest revision**
>
> Summary of changes:
>
> - Improved Table 1 which shows better target-noise separation now across several architectures (ResNet-50 and ResNet-200) and several batch sizes (512 vs 1024), as requested by reviewers.
>
> - Improved Sec. 4 text to describe new experiments, and better highlight our use of significance testing throughout.
>
> - Expanded Discussion section that explains interesting connections between our work and the maximum margin principle that has guided ML for decades.
>
> - New App. N on Corruption analysis. We show that SINCERE and SupCon are indistinguishable (no statistically significant difference) across levels of corruption in the CIFAR-10-C and CIFAR-100-C datasets, thus offering a study of robustness to answer the request by reviewer nwh7.

---

### Decision · Action_Editor_h9TJ · 2025-02-14

**Recommendation:** Reject

**Comment:**

Overall, there are some common issues raised by the reviewers, including
-- The superiority of the proposed method is not well justified, especially, the proposed method does not seem to be empirically better than other baselines such as the supervised contrastive learning.
-- The empirical studies are not sufficient. The proposed method is only tested on some small-scale datasets, which is not representative. The effectiveness of the method needs to be verified on large-scale problems, and on more diverse evaluations in addition to the current transfer learning setting.
-- The presentation and derivations need to be refined. One reviewer pointed out some issues of the theoretical results, which I briefly checked and agreed with most of the comments. I encourage the authors to refine their statements and derivations according to the comments from the reviewer to make the paper more solid.

Based on the above conclusion, I have to propose rejection of the paper. However, I highly encourage the authors to conduct more through evaluations and refine the statements and derivations, which which I believe could be a very good submission for future publication.

**Audience:**

The paper might be beneficial to researchers on self-supervised learning community.

**Claims And Evidence:**

This paper identifies the repelling issue from the same class in supervised conctrastive learning, and proposes a fix by removing data terms from the same class from the negative data in the loss. The method is supported by a probability framework. The proposed method is verified on several standard benchmarks. Unfortunately, some reviewers think the current results do not sufficiently support the claims of the paper. Please see more details in the comment section.

---

> ### Author Response · Authors · 2025-02-15
> **Thanks for your feedback**
>
> Thanks to both action editor and reviewers for the time and effort you've spent in review of our work. We appreciate your feedback, even if it is not the outcome we were hoping for.
>
> I wanted to take a moment to clear up a few misunderstandings.
>
> 1) For **Audience**, we are primarily targeting folks interested in **supervised** contrastive learning.
>
> 2) We did not claim "superiority" of our method in a general sense.
>
> TMLR's main evaluation criteria asks "Are the claims made in the submission supported by accurate, convincing and clear evidence?"
>
> We want to be clear that we never claimed a general sense of "superiority" for our proposed method. Indeed, our paper points out that SINCERE is not statistically any better (but also no worse) than SupCon in terms of accuracy on standard classification tasks. We feel the reviewers are asking us to provide justification for a claim we are not trying to make.
>
> Instead, the *specific* main claims we made throughout the paper remain the following
>
> * SINCERE repairs the intra-class repulsion issue of SupCon, obtaining better target-noise separation
> * SINCERE offers representations that generalize well to new tasks via transfer learning.
>
> Thanks again for your help with our work.